# Concentrations, Particle-Size Distributions, and Dry Deposition Fluxes of Aerosol Trace Elements over the Antarctic Peninsula in Austral Summer

Songyun Fan[1], Yuan Gao[1], Robert M. Sherrell[2], Shun Yu[1], Kaixuan Bu[2]

[1]Department of Earth and Environmental Sciences, Rutgers University, Newark, 07102, USA
[2]Department of Marine and Coastal Sciences, Rutgers University, New Brunswick, 08901, USA

*Correspondence to*: Yuan Gao (yuangaoh@newark.rutgers.edu)

**Abstract.** Size-segregated particulate air samples were collected during the austral summer of 2016-2017 at Palmer Station on the Anvers Island, West Antarctic Peninsula, to characterize trace elements in aerosols. Trace elements in aerosol samples, including Al, P, Ca, Ti, V, Mn, Ni, Cu, Zn, Ce, and Pb, were determined by total digestion and sector field inductively coupled plasma mass spectrometer (SF-ICP-MS). The crustal enrichment factors ($EF_{crust}$) and k-means clustering results of particle size distributions show that these elements are derived primarily from three sources: (1) regional crustal emissions, including possible resuspension of soils containing biogenic P, (2) long-range transport, and (3) sea-salt. Elements derived from crustal sources (Al, P, Ti, V, Mn, Ce) with $EF_{crust}<10$ were dominated by the coarse-mode particles ($>1.8$ µm) and peaked around 4.4 µm in diameter, reflecting the regional contributions. Non-crustal elements (Ca, Ni, Cu, Zn, Pb) showed $EF_{crust}>10$. Aerosol Pb was primarily dominated by fine-mode particles, peaking at 0.14 - 0.25 µm, and likely was impacted by air masses from southern South America based on air-mass back trajectories. However, Ni, Cu, and Zn were not detectable in most size fractions and didn't present clear size patterns. Sea-salt elements (Ca, $Na^+$, $K^+$) showed a single-mode distribution and peaked at 2.5–4.4 µm. The estimated dry deposition fluxes of mineral dust for the austral summer, based on the particle size distributions of Al measured at Palmer Station, ranged from 0.65 to 28 mg m$^{-2}$ yr$^{-1}$ with a mean of $5.5 \pm 5.0$ mg m$^{-2}$ yr$^{-1}$. The estimated dry deposition fluxes of the target trace elements in this study were lower than most fluxes reported previously for coastal Antarctica and suggest that atmospheric input of trace elements through dry deposition processes may play a minor role in determining trace element concentrations in surface seawater over the continental shelf of the West Antarctic Peninsula.

## 1 Introduction

Aerosols affect the climate through direct and indirect radiative forcing (Kaufman et al., 2002). The extent of such forcing depends on both physical and chemical properties of aerosols, including particle size and chemical composition (Pilinis et al., 1995). Size and chemical composition of aerosols influence aerosol optical properties as well as cloud formation and development (Weinzierl et al., 2017), and such information is critically needed in climate model for better estimating aerosol climate effects (Adebiyi and Kok, 2020). In the atmosphere, the removal of aerosols involves gravitational settling, impaction, diffusion, hygroscopic growth, and scavenging by precipitation, and the rates of all these processes are dependent on the aerosol particle size (Saltzman, 2009). Over the Southern Ocean and Antarctica, aerosol particle size distributions have been

studied (Gras, 1995; Järvinen et al., 2013; Xu et al., 2013; Kim et al., 2017; Herenz et al., 2019; Lachlan-Cope et al., 2020). Seasonal variations of the particle number concentrations were observed in the King Sejong Station, Antarctic Peninsula, and Halley Research Station on the Brunt Ice Shelf. The maximum and minimum of the particle number concentrations at two sites

were found in the austral summer and austral winter, respectively (Kim et al., 2017; Lachlan-Cope et al., 2020). Due to the low background concentrations of aerosol particles, new particle formation has been suggested to substantially affect the annual aerosol concentration cycles (Lachlan-Cope et al., 2020). However, most of these studies focused on the physical characterises of aerosol particle size; the size distributions of aerosol trace elements are still poorly understood, and at present only aerosol Fe has been characterized for particle-size distributions around Antarctica (Gao et al., 2013, 2020).

Atmospheric aerosol deposition delivers nutrient elements to the open ocean, playing an essential role in maintaining marine primary production (Jickells and Moore, 2015; Jickells et al., 2016; Mahowald et al., 2018). A significant source of atmospheric trace elements in the remote oceans is continental dust derived from arid and unvegetated regions (Duce and Tindale, 1991). In addition, sea salt emission, volcanic eruptions, biomass burning, anthropogenic activities and even glacial processes contribute trace elements to the atmosphere (Pacyna and Pacyna, 2001; Chuang et al., 2005; Guieu et al., 2005;

Crusius et al., 2011; Baker et al., 2020). The surface concentrations of the trace elements Al, Fe, Mn, Zn, and Pb in several open ocean regions depend strongly on atmospheric inputs (Duce et al., 1991; Prospero et al., 1996; Wu and Boyle, 1997; Measures and Vink, 2000; Moore et al., 2013; Bridgestock et al., 2016).

Atmospheric trace elements over the Southern Ocean and Antarctica may derive from distant continental sources through long-range transport (Li et al., 2008) and local dust sources in certain areas (Kavan et al., 2018; Delmonte et al., 2020). A wide

range of aerosol studies has been carried out in Antarctica, with the intention of understanding the processes affecting aerosols and the background level of trace elements in the atmosphere (Zoller et al., 1974; Dick and Peel, 1985; Tuncel et al., 1989; Artaxo et al., 1990; Lambert et al., 1990; Dick, 1991; Artaxo et al., 1992; Loureiro et al., 1992; Mouri et al., 1997; Mishra et al., 2004; Arimoto et al., 2008; Gao et al., 2013; Xu and Gao, 2014; Winton et al., 2016). In the Antarctic Peninsula, the concentrations of aerosol trace elements were measured at several sites (Dick, 1991; Artaxo et al., 1992; Mishra et al., 2004;

Préndez et al., 2009). Total dust deposition in this region was also estimated based on the ice-core record (McConnell et al., 2007). However, the measurement of particle size distribution of aerosol trace elements in Antarctic Peninsula is missing and there is no direct measurement that evaluated the importance of atmospheric deposition as a source of nutrients for primary producers in West Antarctic Peninsula shelf waters. Over the past several decades, one of the most dramatically warming regions in the Southern Hemisphere has been the Antarctic Peninsula (Vaughan et al., 2003; Bromwich et al., 2013; Turner et

al., 2014). Warming ocean waters have caused most glaciers on the Peninsula to retreat (Cook et al., 2016) and small increases in air temperature are contributing to rapid summer melting and ice loss in this region (Abram et al., 2013).

Under conditions of low precipitation (Van Lipzig et al., 2004) and high wind speed (Orr et al., 2008), several ice-free areas on James Ross Island, off the east coast of the northern Peninsula could serve as local dust sources (Kavan et al., 2018), and such sources may contribute to the atmospheric loading of certain trace elements such as Fe (Winton et al., 2014; Gao et

al., 2020). Similar ice-free areas were found in King George Island, Livingston Island, Anvers Island etc. in the Antarctic

Peninsula region (Bockheim et al., 2013), and may act as potential sources of aeolian dust. As part of the current study, the particle size distribution of aerosol Fe measured at Palmer Station showed single mode distribution and was dominated by coarse particles, suggesting that local regional dust emission dominated the concentration of Fe in this region (Gao et al., 2020). Under the current warming trend, ice core data show that lithogenic dust deposition more than doubled during the 20th century in the Antarctic Peninsula (McConnell et al., 2007). In addition, the rapid warming condition may enhance the emission of other sources. For example, the Antarctic Peninsula has been suggested as one of the sites that have the highest P excretions contributed by seabird colonies globally (Otero et al., 2018). The enhanced local dust emission may thus cause an increased P emission as well. Consequently, dust emissions induced by regional warming may have impacted the concentrations of aerosol trace elements in the marine atmosphere over the Antarctic Peninsula, affecting atmospheric deposition of trace elements to coastal waters off the Antarctic Peninsula and adjacent pelagic waters of the Southern Ocean (Wagener et al., 2008). However, aerosol trace elements are still under-sampled around coastal Antarctica, and thus quantification of the chemical and physical properties of aerosols and accurate estimation of the atmospheric deposition of trace elements to the region are inadequate.

This study presents multi-element results from an *in-situ* measurement of size-segregated aerosol particles at Palmer Station, Antarctic Peninsula, in the austral summer of 2016–2017. The objectives are to (1) measure the concentrations and size distributions of a suite of aerosol trace elements, (2) determine potential sources of the elements, and (3) estimate dry deposition fluxes of the elements based on the concentrations in ten size classes of aerosols. Results from this study fill a data gap critically needed for characterization of aerosol properties and for improving quantification of the fluxes that contribute to regional biogeochemical cycles. The new observational data also provide insight into sources of aerosol trace elements, as influenced recently by warming, which exposes a greater area of ice-free land, and by the impact of human activities in this region. A full discussion of atmospheric Fe in this sample set was published recently (Gao et al., 2020); this paper extends that study by investigating the concentrations, size distributions, and dry deposition fluxes of a suite of additional aerosol trace elements.

## 2 Methods

### 2.1 Sampling and sample treatment

Aerosol size-segregated samples were collected during austral summer from November 19, 2016 to January 30, 2017 at Palmer Station (64.77° S, 64.05° W, Figure 1), located on the southwestern coast of Anvers Island off the Antarctica Peninsula. A detailed description of the aerosol sampling, including the protocols for mitigating contamination in the pristine environmental can be found in Gao et al. (2020). Briefly, sampling was conducted using a ten-stage Micro-Orifice Uniform Deposit Impactor[TM] (MOUDI, MSP Corp., MN, USA) with a 30 L min[-1] flow rate. The 50% cut-off aerodynamic diameters of MOUDI are 0.056, 0.10, 0.18, 0.32, 0.56, 1.0, 1.8, 3.2, 5.6, 10, and 18 μm. In this study, size fractions ≤1.0 μm were summed to operationally define fine-mode particles and those ≥1.8 μm were summed to define coarse-mode particles, similar to previous studies which operationally divided aerosol particles into fine and coarse fractions using a cut-off size of 1.0–3.0 μm (Siefert

et al., 1999; Chen and Siefert, 2004; Buck et al., 2010; Gao et al., 2019). The aerosol sampler was placed on a sampling platform which was ~300 m east from the station center and ~3 m high above the ground (~20 m above sea-level) in "Palmer's backyard" (Gao et al., 2020). To avoid local contamination from the research station, a wind control system was set up to pause aerosol sampling when wind direction was inside the sector ±60° from the direction of the station buildings or when wind speed <2 m s$^{-1}$. The active sampling time was about 71-98% of the total sampling time (Table 1). Due to extremely low concentrations of aerosol trace elements over Antarctica, the duration of each sampling event was approximately one week (Table 1).

After each sampling, the MOUDI sampler was carried back to the lab in the research station for sample filter changing and sampler cleaning in a Class 100 cleanroom flow bench. Aerosol samples were stored frozen in pre-cleaned Petri dishes at -20°C before analyses. A total of 8 sets size-segregated aerosol samples were collected on Teflon filters (1 µm pore size, 47 mm diameter, Pall Corp., NY, USA). A full set of blank filters (n = 11) was mounted on the sampler, carried to the sampling platform without running the sampler, and thus defined as field blanks. Meteorological conditions were recorded *in situ* by a weather station (Campbell Scientific, UT, USA) installed on the same platform (Table 1).

## 2.2 Chemical analyses

### 2.2.1 Trace elements in aerosols

Aerosol samples were analyzed for the concentrations of trace elements by an Element-1 sector field inductively coupled plasma mass spectrometer (SF-ICP-MS, Thermo-Finnigan, Bremen, Germany) at the Department of Marine and Coastal Sciences of Rutgers University, following a strong-acid digestion method described in Gao et al. (2020). Elemental concentrations were determined for Al, P, Ca, Ti, V, Mn, Ni, Cu, Zn, Ce, and Pb. Briefly, a quarter of each sample filter was digested in a closed 15 mL Teflon vial (Savillex, MN, USA) with Optima Grade HF (0.1 mL) and HNO$_3$ (0.8 mL) (Fisher Scientific, NJ, USA). Sample digestion was performed on a uniform heating HPX-200 (Savillex, MN, USA) hot plate for 4 hours at 165 °C followed by complete evaporation of acids. Then, 2.0 mL 3% HNO$_3$ with 1 ppb Indium (In) solution was added to re-dissolve the sample, the In used as an internal standard to correct instrument drift in the ICP-MS analyses. All the digestion processes were carried out in a HEPA filter-controlled Class 100 clean hood in the Atmospheric Chemistry laboratory at Rutgers University. The Teflon vials and test tubes used in this study were thoroughly acid-cleaned. To ensure the data quality, for each batch of samples, at least two procedural blanks were processed in the same way as the samples to monitor for possible contamination. During the ICP-MS analysis, duplicate injections of sample solutions were made every ten samples to check the instrument precision (Table S1). The recovery of this analytical protocol was estimated by 7 separate digestions of the Standard Reference Material (SRM) 1648a-urban particulate matter (National Institute of Standard and Technology, MD, USA) (Table S1). The method limits of detection (LOD) were calculated as three times the standard deviation of 11 field blanks and a 200 m$^3$ representative sampling volume (Table S1). The medians of %blank in samples for detectable trace elements were calculated for quality control (Table S1). The concentrations below LOD were given a concentration of zero

for the purposes of this study. Elements with all concentrations lower than the LOD, including Cr, Co, Cd and Sb, were measured but are not reported or discussed. The aerosol Fe concentrations were reported in Gao et al. (2020) and are not included in this paper. Although the fractional solubility of Fe was obtained, the solubilities for the other trace elements are not measured or discussed in this study.

### 2.2.2 Ionic tracers in aerosols

The concentrations of water-soluble $Na^+$ and $K^+$ in aerosols were analyzed by ion chromatography (IC) (ICS-2000, Dionex, CA, USA) with an IonPac CS12A ($2 \times 250$ mm$^2$) analytical column at the Atmospheric Chemistry laboratory at Rutgers University. The cations $Na^+$ and $K^+$ were used as tracers to estimate the portion of aerosols derived from seawater and biomass burning, respectively. The non-sea-salt $K^+$ (nss-$K^+$) was estimated using the equation: [nss-$K^+$] = [$K^+$] - 0.037[$Na^+$]. Sample processing for IC analysis was similar to the method used by Zhao and Gao (2008) and Xu et al. (2013). Briefly, a quarter of

the sample filter was transferred to a plastic test tube and leached with 5.0 mL Milli-Q water in an ultrasonic bath for 20 minutes at room temperature. Before being injected into the IC, the leachate was filtered through a PTFE syringe filter (0.45 µm pore size, VWR, PA, USA). The method LODs for $Na^+$ and $K^+$ based on 7 blanks and a 200 m$^3$ representative sampling volume were 2 and 1 ng m$^{-3}$, respectively. The precision of the analytical procedures based on seven spiked samples was $<\pm1\%$.

## 2.3 Data analyses

### 2.3.1 Enrichment factors

To achieve an initial estimate of the possible sources for trace elements, enrichment factors relative to upper continental crust (EF$_{crust}$) were calculated, using the equation:

$$EF_{crust} = \frac{(X_i/X_{Al})_{sample}}{(X_i/X_{Al})_{crust}} \quad (1)$$

where (Xi/X$_{Al}$)$_{sample}$ is the mass concentration ratio of element i to the crustal reference element, Al, in aerosol samples, and (Xi/X$_{Al}$)$_{crust}$ is the abundance ratio of element i to Al in the upper continental crust (Taylor and McLennan, 1995). The crustal reference element Al has been widely used to calculate crustal enrichment factors in the Southern Ocean and Antarctica (Zoller et al., 1974; Dick, 1991; Lowenthal et al., 2000; Xu and Gao, 2014). When the EF$_{crust}$ is greater than 10, the element likely has additional contributions from other sources (Weller et al., 2008).

### 2.3.2 Particle size distribution and k-means clustering

The aerosol particle size distributions were converted to normalized concentrations which is defined as the concentration of trace elements in a size bin divided by the width of the bin (Warneck, 1988):

$$\frac{dC}{d\log(D_p)} = \frac{dC}{\log(D_{p,u}) - \log(D_{p,l})}$$

In the equation, $dC$ is the mass concentration of trace elements in a size bin, and $d\log(D_p)$ is the difference in the log of the bin width. The $d\log(D_p)$ is calculated by subtracting the log of the lower bin boundary ($\log(D_{p,u})$) from the log of the upper bin boundary ($\log(D_{p,l})$). The k-means clustering algorithm was used to cluster the average particle size distribution of each trace element. The optimal number of clusters (k) was selected by choosing the k with the highest Calinski-Harabasz index (Caliński and Harabasz, 1974).

### 2.3.3 Atmospheric dry deposition flux estimation

Dry deposition flux ($F_d$, mg m$^{-2}$ yr$^{-1}$) of each element in aerosols was calculated from the concentration ($C_e$, ng m$^{-3}$) of the trace element in the air and dry deposition velocity ($V_d$, cm s$^{-1}$):

$$F_d = 0.315 \times V_d \times C_e \quad (2)$$

where 0.315 is a unit conversion factor (Gao et al., 2013). The $V_d$ for each trace element was computed by dry deposition rates ($V_{di}$, cm s$^{-1}$) and particle distribution ratios ($P_{dri}$, %) following the equation:

$$\sum_{i=1}^{10} V_{di} \times P_{dri} \quad (3)$$

The $P_{dri}$ was derived from the concentrations of trace elements in different size fractions i and $V_{di}$ was estimated using a combination model of Williams (1982) and Quinn and Ondov (1998). This model includes the effects of wind speed, air/water temperature difference, sea surface roughness, spray formation in high wind speed and relative humidity. Meteorological parameters used for estimating dry deposition rates were measured *in situ* by the weather station with 1 minute temporal resolution that was converted to 1 hour averages. Sea surface temperature data were obtained from the Palmer Station Long-Term Ecological Research (LTER) study (https://oceaninformatics.ucsd.edu/datazoo/catalogs/pallter/datasets/28). Dry deposition rates of coarse-mode particles were dominated by gravitational settling, whereas the dry deposition rates of smaller particles were controlled by environmental factors, including wind speed, relative humidity, air temperature, and sea surface temperature (Figure 2). Therefore, the uncertainties of the dry deposition velocity estimation for the elements that were dominated by coarse particles were about ±30-60%, and the uncertainty of fine particle dominated element (Pb) was about ±100%. The overall estimation of dry deposition flux usually carries substantial uncertainty (a factor of 2 to 4) due to the limited sampling volume and the assumptions inherent to the $V_{di}$ estimation (Duce et al., 1991; Gao et al., 2020). The dry deposition velocities determined for each element are reported in Table 3. Dry deposition fluxes were calculated for the trace elements showing clear particle size distribution patterns: Al, P, Ca, Ti, V, Mn, Ce, and Pb. For Ni, Cu, and Zn, that didn't show clear size distributions, the ranges of their dry deposition fluxes were also estimated by applying their mean concentrations to the lowest (Pb, 0.11 ± 0.12 cm s$^{-1}$) and highest (Al, 0.49 ± 0.28 cm s$^{-1}$) dry deposition velocities. The dry deposition fluxes of dust were estimated based on the concentrations and particle size distributions of Al in aerosols, assuming that Al accounted for 8% of dust mass (Taylor and McLennan, 1995).

### 2.3.4 Air mass back trajectories

To explore possible source regions of air masses affecting trace elements in aerosols collected at Palmer Station, the NOAA Hybrid Single Particle Lagrangian Integrated Trajectory Model (HYSPLIT) was used to calculate 72-hour air mass back trajectories for each sampling duration (Rolph et al., 2017). In this study, the HYSPLIT model was driven by the meteorological data from the Global Data Assimilation System (GDAS) with a 0.5-degree resolution. Each air mass back trajectory was calculated at 3-hour intervals and started from one-half mixed boundary layer height. The back trajectories during each

sampling period were used to calculate trajectory frequencies which were defined by the following equation:

$$\text{Trajectory frequencies} = 100 \times \frac{\text{number of endpoints per grid square}}{\text{number of trajectories}} \quad (4)$$

### 3 Results and discussion

### 3.1 Enrichment factors of trace elements

Crustal enrichment factors of trace elements in aerosols were calculated as the first step of source identification (Figure 3).

Two major EF groups were found, representing crustal and non-crustal elements as follows.

### 3.1.1 Crustal elements (P, Ti, V, Mn, Ce)

The values of $EF_{crust}$ for Ti, V, Mn, and Ce in aerosol samples were less than 10, indicating that a crustal source is the dominant source for those elements (Figure 3). As typical lithogenic elements (Boës et al., 2011), Ti and Mn in aerosols over oceanic regions are usually derived from natural dust emissions (Shelley et al., 2015; Marsay et al., 2018; Buck et al., 2019). Likewise,

V and Ce in aerosols over the South Pole were reported dominated by crustal weathering (Zoller et al., 1974). In addition to crustal emissions, long-range transport may deliver some portion of these trace elements from remote sources to Antarctica (Wagener et al., 2008). For example, aerosol Ce and Mn derived from anthropogenic emissions were thought to be contributed by additives in vehicle fuels (Fomba et al., 2013; Gantt et al., 2014), and V in aerosols was found associated with ship emissions due to the use of heavy oil fuel (Keywood et al., 2020). However, the $EF_{crust}$ results from this study suggest that nearby fuel

combustion did not cause significant enrichment of V in aerosols at Palmer Station. Similarly, unenriched V was observed at McMurdo Station where light-weight fuel oil was used that was not a significant source of V (Lowenthal et al., 2000). We conclude that Ti, Mn, V, and Ce observed at Palmer Station were derived primarily from crustal sources.

  The range of $EF_{crust}$ for P was between 2 and 8, relatively higher than that of the other crustal elements. In Antarctic soils, P has been widely studied (Campbell and Claridge, 1987; Blecker et al., 2006; Prietzel et al., 2019). Around the Antarctic

coast, including the northern end of the Antarctic Peninsula, high P inputs to the surface soil were found in seabird colonies (Otero et al., 2018), and a high enrichment of P ($EF_{crust}$ = 33) was reported previously at King George Island (Artaxo et al., 1990). The closest potential source, the penguin colony on Torgerson Island, is only about 1 km from Palmer Station. Given that regional wind-induced dust likely affects aerosol composition over the Antarctic Peninsula (Asmi et al., 2018; Gao et al.,

2020), soil-derived P is likely to be emitted to the atmosphere. In addition, biogenic activities in Antarctica also produce abundant gaseous P, such as phosphine, through anaerobic microbial processes in soils and animal digestives, and phosphine gas can be transformed to other low-volatile P-containing compounds in the atmosphere or soils (Zhu et al., 2006). Primary biogenic aerosols, sea-salt aerosols, and volcanic emissions could also contribute P to the atmosphere, causing an elevated $EF_{crust}$ for P (Zhao et al., 2015; Trabelsi et al., 2016).

### 3.1.2 Non-crustal elements (Ca, Ni, Cu, Zn, Pb)

The enrichment factors of atmospheric Ni, Cu, Zn, and Pb relative to the crustal element Al, were found to be greater than 10 in some samples, suggesting contributions from non-crustal sources during the corresponding sampling periods in this study (Figure 3). High enrichments of these elements in Antarctica have commonly been associated with long-range transport derived from anthropogenic emission (Boutron and Lorius, 1979; Maenhaut et al., 1979; Dick, 1991; Artaxo et al., 1992; McConnell et al., 2014; Xu and Gao, 2014). Aerosol Cu, Zn, and Pb are contributed primarily by combustion or industrial activities (Pacyna and Pacyna, 2001). Strong variations of the $EF_{crust}$ of Cu, Zn, and Pb observed in snow samples at Dome C, Antarctica, over the 20th century, were attributed to volcanic activities (Boutron and Lorius, 1979). On the other hand, heavy oil combustion was found to be the major source of aerosol Ni and V, and V/Ni ratios are usually used to identify shipping emissions (Keywood et al., 2020). Nevertheless, the V/Ni measured at Palmer Station ranged from 0.01 to 0.2, much lower than the V/Ni = 3.2 ± 0.8 characteristic of the discharge from ship engines (Viana et al., 2009; Viana et al., 2014; Celo et al., 2015). Hence, despite the recent increase in tourist ship traffic (Lynch et al., 2010), it seems that Palmer Station was barely impacted by ship emissions, which is consistent with the $EF_{crust}$ of V. As a common crustal element, Ca accounts for about 3.5% of the weight of Earth's crust (Taylor and McLennan, 1995), while Ca is also a conservative major ion in seawater (Millero, 2016). The high $EF_{crust}$ for Ca at Palmer Station suggests that aerosol Ca was mainly derived from sea-salt.

### 3.2 Concentrations of trace elements

The concentrations of Ca and Al, the two major elements measured in this study, were one to several orders of magnitude higher than other elements (Table 1). To place all of the results of this study into the context of past investigations, we provide a visual comparison of measured concentrations of aerosol trace elements over Antarctica (Figure 4).

### 3.2.1 Mineral dust (Al, P, Ti, V, Mn, Ce)

The concentrations of Al in aerosols varied from 1.2 to 7.9 ng m$^{-3}$ with an average of 4.3 ng m$^{-3}$ during 2016-2017 austral summer in this study. These concentrations are lower than the 4-year mean Al values of ~13 ng m$^{-3}$ at King George Island (62° S, 58° W), at the northern end of the Antarctic Peninsula (Artaxo et al., 1992), but were slightly higher than the 2-year mean Al concentrations of 1.9 ng m$^{-3}$ observed at King Sejong Station (62° S, 59° W) (Mishra et al., 2004), the summer mean of 0.194 ng m$^{-3}$ at Larson Ice Shelf to the southeast of Palmer Station (Dick et al., 1991), the 5-year summer mean of 1.3 ng m$^{-3}$

in East Antarctica (Weller et al., 2008) and the summer average of 0.57 ng m$^{-3}$ measured at the South Pole (Zoller et al., 1974;

Maenhaut et al., 1979) (Figure 4b). All these results, including ours, were much lower than the average Al concentrations of 180 ng m$^{-3}$ and 250 ng m$^{-3}$ observed on the two sites at McMurdo Station (Mazzera et al., 2001) due to the impact of the McMurdo Dry Valleys (Figure 4b). The nearby McMurdo Sound was reported as the dustiest site in Antarctica (Winton et al., 2016). These aerosol Al concentrations are also lower than the average Al concentration 190 ng m$^{-3}$ observed in coastal East Antarctica where the samples were also impacted by air masses passing over McMurdo regions (Xu and Gao, 2014) (Figure

4b). The concentrations of Ti ranged from 140 to 800 pg m$^{-3}$ with an average of 250 pg m$^{-3}$, while the concentration of Mn ranged from 17 to 44 pg m$^{-3}$ with an average of 30 pg m$^{-3}$. Both Ti and Mn concentrations at Palmer Station were lower than yearly mean concentrations at King George Island (average Ti: 1600 pg m$^{-3}$, average Mn: 660 pg m$^{-3}$) (Artaxo et al., 1992), and they were also lower than the summer mean PM$_{10}$ concentrations at McMurdo Station (Ti: 26000 pg m$^{-3}$, Mn: 3000 pg m$^{-3}$) (Mazzera et al., 2001). However, these Ti and Mn values observed at Palmer Station were in the same magnitude with the

PM$_{2.5}$ concentrations during the austral summer near Chilean Bernardo O'Higgins base, located on the northwest coast of the Antarctic Peninsula (Préndez et al., 2009) and with the concentrations in bulk aerosols over coastal East Antarctica in austral summer (Mn: 450–1200 pg m$^{-3}$ with an average of 700 pg m$^{-3}$) (Xu and Gao, 2014) (Figure 4e and g). The concentrations of V ranged from 2.7 to 6.1 pg m$^{-3}$ with a mean value of 4.2 pg m$^{-3}$, which is higher than the numbers reported at the South Pole (average ~1.5 pg m$^{-3}$) (Zoller et al., 1974; Maenhaut et al., 1979) (Figure 4f). However, the V concentration observed at Palmer

Station was much lower than previous observations in North Atlantic Ocean (50–3170 pg m$^{-3}$) (Fomba et al., 2013) and eastern Pacific Ocean (average 150 pg m$^{-3}$) (Buck et al., 2019). On the other hand, Ce demonstrated an average concentration of 1.3 ± 0.69 pg m$^{-3}$, which is consistent with the Ce concentrations reported at Neumayer Station, Antarctica (Weller et al., 2008) (Figure 4k). The low concentrations of aerosol V and Ce suggest that Palmer Station was not significantly influenced by fossil fuel combustion. The P concentrations in aerosols during this study ranged from 85 to 250 pg m$^{-3}$ with an average of 150 ng

m$^{-3}$. The concentrations of aerosol P at Palmer Station were lower than that at King George Island (3820 pg m$^{-3}$) (Artaxo et al., 1990). Comparing global aerosol P concentrations, we find that P concentrations over the Antarctic Peninsula are in the same range as those over the Central Pacific Ocean (Chen, 2004). In this study, nss-K$^+$ was used as a tracer of biomass burning (Winton et al., 2015). The calculated nss-K$^+$ was indistinguishable from zero, suggesting that K$^+$ in aerosol at Palmer Station was primarily derived from sea water, not from biomass burning through long-range transport. These results agree well with

the air mass back trajectories, which indicate that most samples collected during this study were barely affected by South America (Figure 5). In addition, for most of the 72-hour air-mass back trajectories, the highest frequencies were found around the northern Antarctic Peninsula, suggesting that aerosol crustal elements observed at Palmer Station were impacted by sources in that region (Figure 5).

### 3.2.2 Anthropogenic aerosols (Ni, Cu, Zn, Pb)

The concentrations of Ni, Cu, Zn, and Pb in were more variable than the crustal elements. High levels of these elements would suggest an effect of a polluted air mass derived from long-transport or regional air pollution (Artaxo et al., 1992). The highest

concentration of Ni observed during this study was 320 pg m$^{-3}$, with an average of 75 pg m$^{-3}$, while the lowest Ni concentration was below LOD (<20 pg m$^{-3}$). Samples M2 and M10 showed relatively high values of Ni with 320 and 200 pg m$^{-3}$, respectively, while the concentrations of Ni in M1, M5, and M6 were much lower, ranging from 17 to 37 pg m$^{-3}$. The concentrations of Cu varied from <20–480 pg m$^{-3}$ (average 150 pg m$^{-3}$), while Zn ranged from <30–710 pg m$^{-3}$ (average 200 pg m$^{-3}$). These results indicate that the concentrations of these elements in aerosols varied dramatically throughout the study period, likely affected by the source strength and meteorological conditions. From the air mass back trajectories, the samples with high Ni (M2, M10), Cu (M1, M2, M4, M10) and Zn (M5) all were impacted by significant amount of air masses from the South Pacific Ocean and South America (Figure 5). The back trajectories of air masses of M7 didn't touch South America but the concentration of Zn in this sample was high. With the fact that aerosol Zn was found in both fine- and coarse-mode fractions (Table 2), both local sources and long-range transport may contribute to this element in the air. Artaxo et al. (1992) collected aerosol samples using a 2-stage size-segregated sampler and observed high concentrations of Ni, Cu, and Zn with averages of 240, 790 and 7200 pg m$^{-3}$ at King George Island in summertime (Figure 4h, i and j), much higher than the results from this study. The high concentrations may have resulted from local polluted dust emissions around their sampling site (Hong et al., 1999). In addition, their sampling site was considerably north of Palmer Station, and may be more likely to be impacted by air masses from South America (Chambers et al., 2014). In South America, high enrichment of Ni, Cu, Zn, and Pb in fine-mode particles was reported to be primarily associated with vehicle emission, soil dust, and oil combustion (Artaxo et al., 1999; Jasan et al., 2009). Moreover, miming activities were suggested as an important source, especially in remote sites in South America (Carrasco and Préndez, 1991; Klumpp et al., 2000). On the other hand, the concentrations of Ni, Cu, and Zn at Palmer Station were lower than some other sites in Antarctica. At McMurdo Station, the average concentrations of Cu and Zn in PM$_{10}$ samples were 200 and 1200 pg m$^{-3}$ (Mazzera et al., 2001) (Figure 4i and j). Over coastal East Antarctica, the Ni concentrations in aerosols ranged from <3 to 2200 pg m$^{-3}$ with an average of 750 pg m$^{-3}$ (Xu and Gao, 2014) (Figure 4h).

The concentrations of aerosol Pb observed at Palmer Station ranged from 5.0 to 60 pg m$^{-3}$ with an average of 19 pg m$^{-3}$, higher than the average concentration of 4.7 pg m$^{-3}$ previously observed on the east coast of the Antarctic Peninsula in austral summer (Dick, 1991) (Figure 4l) and the annual average concentration of 11 pg m$^{-3}$ in Tasmania (Bollhöfer et al., 2005). However, the average Pb concentration reported at King George Island was about 800 pg m$^{-3}$ (Artaxo et al., 1992), considerably higher than observed in this study (Figure 4l). The two highest Pb concentrations observed during this study were in M4 (60 pg m$^{-3}$) and M5 (30 pg m$^{-3}$). The results of air mass back trajectories for M4 and M5 show that the air masses were derived in part from South America, which suggests additional contribution from long-range transport (Figure 5b and c). However, when the 72-hour air mass back trajectories did not intersect the South American continent, the concentrations of Pb in those samples were significantly lower (Figure 5a and d). The low concentrations of Pb observed in samples associated with air masses that did not pass over Southern South America suggest that local anthropogenic emissions were negligible. Thus the major source of non-crustal elements in aerosols over the study region may be long-range transport of aerosols from Southern Hemispheric continental regions containing a mixture of anthropogenic and crustal emissions.

### 3.2.3 Sea salt aerosol (Ca, Na$^+$, K$^+$)

The concentrations of sea-salt species were much higher than the other trace elements (Table 1). The concentrations of aerosol Ca were the highest among all elements measured in this study (range 16–53 ng m$^{-3}$ and mean value of 30 ng m$^{-3}$). The mean value of Ca is close to the mean Ca concentrations observed previously in coastal regions of the Antarctic Peninsula (Artaxo et al., 1992; Weller et al., 2008) (Figure 4d). In addition, Na$^+$, as a tracer of sea-salt, and K$^+$, as a tracer of biomass burning (Zhu et al., 2015), were used to further evaluate the contribution of trace elements from other sources. The concentrations of Na$^+$ and K$^+$ showed average concentrations of $890 \pm 310$ and $28 \pm 11$ ng m$^{-3}$, respectively. Thus, the Na$^+$/K$^+$ ($32 \pm 3.5$) ratios was close to the average Na/K mass ratio in seawater (27) and the Na$^+$/Ca ratios ($31 \pm 5.5$) were close to the average Na$^+$/Ca ratio in seawater (26) as well (Millero, 2016). Such results agree well with the Na/K (29.7) and Na/Ca (39.9) ratios measured in snow samples collected at James Ross Island (Aristarain et al., 1982). Therefore, the Ca and K$^+$ in aerosols were derived primarily from sea salt at Palmer Station.

### 3.3 Aerosol particle-size distributions of trace elements

The concentrations of trace elements in aerosols observed during this study varied as a function of particle size. We applied k-means clustering to the full data set, and the results indicate that aerosol trace elements in the austral summer at Palmer Station can be classified into five groups based on their normalized particle size distributions (Figure 6), with each group showing a unique size distribution pattern: (1) crustal elements from crustal weathering and wind-induced resuspension of soil particles, (2) Al dominated by local minerals, (3) Pb from anthropogenic sources, as a result of long-range transport, (4) sea salt elements from the ocean, through bursting bubbles of seawater, and (5) P from local biogenic and soil resuspension. The particle size distribution of each element can be classified as single mode, bimodal, or trimodal, which hints that the elements derive from distinct sources or experienced different processes during the transport to the sampling site.

### 3.3.1 Single mode distribution

The particle size distributions of Na$^+$, P, K$^+$, Ca, Ti, and Ce showed a clear single coarse-mode distribution (Figure 7). As conservative species in seawater, Na$^+$, K$^+$, and Ca, were likely dominated by sea salt aerosols and all had a coarse primary mode at approximately 2.5 - 4.4 µm. Such a pattern agrees well with the particle size distributions of sea salt aerosols measured in coastal East Antarctica (Xu et al., 2013). Similarly, Ti and Ce derived from crustal emission were dominated by coarse-mode particles, peaking at around 4.4 µm. The aerosol P had a slightly shifted single mode distribution and peaked between the coarse-mode crustal aerosol and the sea salt aerosol at around 2.5 - 4.4 µm. Consequently, P was likely controlled by a regional emission source, such as soil resuspension. Given that gravitational settling greatly limits the residence time of coarse-mode particles compared with fine-mode particles, the elements with single coarse-mode distribution are likely to have a relatively proximal dominant source, such as the ice-free areas in the Antarctic Peninsula during austral summer.

### 3.3.2 Bimodal distribution

Although V and Mn also had a single mode distribution in most samples, bimodal distribution was found in a few samples, hinting at contributions from multiple sources (Figure 8). The particle size distribution of V and Mn primarily peaked around 4.4 µm while a small fine-mode peak could be seen around 0.14 - 0.25 µm in sample M2 and M4 for V, and a similar fine-mode peak also presented in M4 for Mn. A different Mn particle size distribution was dominated by particles larger than 18 µm, which is similar to Al and will be discussed below in section 3.3.3. As we concluded above, V and Mn were dominated by crustal emission and the coarse primary mode indicates regional crustal sources. As the size of dust particles decreases with distance away from the sources due to the higher deposition rates of coarse particles (Duce et al., 1991; Baker and Jickells, 2006). The fine-mode peaks are likely caused by additional input from long-range transport. The particle size distribution of V in M2 showed a unique peak in the fine-mode range. In addition, M2 had the highest concentrations of Ni and Cu among all the samples. The air mass back trajectories suggested a contribution from the South Pacific Ocean and coastal South America. This suggests that the fine-mode particles in M2 might be influenced by the remote anthropogenic emissions. By contrast, the air mass back trajectories for M4 directly passed the South American continent. An elevated Pb concentration and fine-mode peaks in Al and Pb size distributions were observed also in M4. Such results suggest M4 may have received polluted dust derived from South America.

### 3.3.3 Trimodal distribution

Aerosol Pb showed bimodal and trimodal distributions in this study (Figure 8). Comparing with the other elements, the primary mode of Pb was much finer and peaked at around 0.14 - 0.25 µm. The primary fine-mode fractions of aerosol Pb in some samples suggest that aerosol Pb at Palmer Station was occasionally dominated by anthropogenic sources as Pb is usually concentrated in fine particles from industrial or traffic emissions (Mamun et al., 2020). Air mass back trajectories also suggest the high Pb samples (M4, M5) received air mass derived from southern South America through long-range transport. The secondary mode of Pb was found around the same size as the crustal elements (approximately 4.4 µm), indicating a possible regional emission, including both crustal sources and local contaminated soil (Santos et al., 2005). In addition, a tertiary mode was observed in M4 and peaked around 0.78 - 1.4 µm. Considering M4 was influenced by long-range transport, Pb at this size range may be dominated by distinct remote sources. Thus Pb at Palmer Station was largely controlled by remote sources in South America. Because Pb was dominated by fine particles, an extremely small dry deposition velocity would be expected (Table 2). This might explain why previous studies in the Antarctic Peninsula suggested aerosol Pb was contributed by remote anthropogenic emission (Dick, 1991), whereas Pb in snow appeared to be associated with natural aerosols (Suttie and Wolff, 1992).

The average Al particle size distribution showed trimodal distribution with a primary mode at particle size larger than 18 µm, a secondary mode around 4.4 µm, and a tertiary mode around 0.14 - 0.25 µm (Figure 8). A single-mode Al particle size distribution peaked at around 4.4 µm which shared the same size with the primary mode of Ti, V, Mn, and Ce in sample M8

and M9, indicating the regional crustal emissions. Additional high peak (> 18 µm) was found in M1, M2, and M7 that contributed to bimodal distribution. A similar distribution can be seen in the Mn distribution of M2. We speculate that high Al concentrations associated with large particles might be caused by local soil resuspension around the sampling site, introducing
large Al-containing particles that may come from aluminosilicate minerals. Such large Al-containing particles may not be unusual, as large-size dust particles were found in Antarctic snow (Winton et al., 2016). Trimodal distribution involved another mode in fine particles, approximately 0.14 - 0.25 µm, in M4 and M5. The high fine-mode Al of M4 and M5 likely derives from long-range transport from South America, which matches with the two highest Pb concentrations in this study.

### 3.3.4 Particle size distribution of mineral dust in Antarctica

Although the particle size distributions of crustal elements (Al, Ti, V, Mn, Ce) have been reported only rarely for Antarctica, the particle size distributions of mineral dust in aerosols, snow, and ice core samples were studied at several sites. The crustal aerosols sampled at Aboa research station, East Antarctica, had a single mode distribution peaking at particle size larger than 8.5 µm (Virkkula et al., 2006). In addition, the grain size distribution of dust particles in snow collected at Berkner Island was largely dominated by particles larger than 5 µm, suggesting a significantly higher proportion of dust was in coarse-mode than
the previous size distribution observed at Kohnen Station, East Antarctic Plateau (Bory et al., 2010). Similarly, at Roosevelt Island, the particle size distributions of dust particles in snow were primarily dominated by coarse particles, possibly larger than 9 µm (Winton et al., 2016). Likewise, the dust particle size distribution in Dome C during the Last Glacial Maximum and the Holocene showed single-mode and peaked at approximately 2 - 3 µm in ice core samples (~3233 m above sea level) (Delmonte et al., 2002; Potenza et al., 2016). Therefore, the dust particle size distributions measured in coastal Antarctic
regions are usually concentrated in coarse particles, whereas the dust particle size distributions of inland Antarctic regions, with longer pathways from the sources, show peaks in relatively finer particles.

### 3.4 Atmospheric dry deposition fluxes

Aerosol particle size distributions have been used to estimate atmospheric dry deposition fluxes of aerosol Fe to the Southern Ocean and coastal Antarctica (Gao et al., 2013; 2020). Similarly, the dry deposition fluxes of aerosol trace elements were
estimated in this study (Table 3). Given that the sea-salt elements were the most abundant, Ca showed the highest dry deposition flux which varied from 0.59 to 6.6 mg m$^{-2}$ yr$^{-1}$ with an average of $1.2 \pm 0.7$ mg m$^{-2}$ yr$^{-1}$. At the other extreme, due to low concentration and fine-mode particle dominance, aerosol Pb showed ~5000-fold lower dry deposition fluxes with a mean of $0.30 \pm 0.23$ µg m$^{-2}$ yr$^{-1}$. The rough estimates of the dry deposition fluxes of Ni and Zn at Palmer Station are close to the median deposition fluxes found in the western North Atlantic Ocean (Ni: 18 µg m$^{-2}$ yr$^{-1}$, Zn: 16 µg m$^{-2}$ yr$^{-1}$), whereas the
dry deposition flux of Cu is slightly higher than the median Zn dry deposition flux (2.8 µg m$^{-2}$ yr$^{-1}$) in the western North Atlantic Ocean (Shelley et al., 2017). However, the dry deposition flux of Cu at Palmer Station is in the lower range of that observed in the eastern North Atlantic Ocean, 3.97 - 194 µg m$^{-2}$ yr$^{-1}$ (Buck et al., 2019). The estimated dry deposition fluxes of total continental dust at Palmer station ranged from 0.65 to 28 mg m$^{-2}$ yr$^{-1}$ with a mean of $5.5 \pm 5.1$ mg m$^{-2}$ yr$^{-1}$, and this

fluxes is only around 10% of the mean global dust deposition flux at remote sites (Lawrence and Neff, 2009). Previous
modelling studies estimated that wet deposition of dust through precipitation scavenging accounted for about 40~60% of the
total deposition in the coastal and open oceans in the mid and low latitude oceans (Gao et al., 2003), and a similar wet deposition
fraction was found in the Southern Ocean and coastal East Antarctica (Gao et al., 2013). Although precipitation varies in
different regions, this is the best estimate we have for Antarctic regions. Assuming this wet deposition fraction (0.4-0.6) applies
to the Antarctic Peninsula region, we approximate roughly a total dust flux of $10 \pm 10$ mg m$^{-2}$ yr$^{-1}$. This value is at the lower
end of the dust flux range, ~5 to ~50 mg m$^{-2}$ yr$^{-1}$, estimated from ice core measurements at James Ross Island on the Antarctic
Peninsula over the 20th century (McConnell et al., 2007), but it is higher than dust deposition flux at Dome C (0.2–0.6 mg m$^{-2}$ yr$^{-1}$) (Lambert et al., 2012), at Talos Dome (0.70 - 7.24 mg m$^{-2}$ yr$^{-1}$) (Albani et al., 2012) and the model estimate for this
region (1.8–3.7 mg m$^{-2}$ yr$^{-1}$) (Wagener et al., 2008). Considering active dust sources were reported at James Ross Island (Kavan
et al., 2018), sites close to James Ross Island are likely to receive higher dust deposition fluxes. In addition, without
measurements of wet deposition, the total dust deposition flux in our study could be underestimated. Comparing with the
global data, the dust deposition flux at Palmer Station is far lower than the total dust deposition fluxes in the North and South
Atlantic Ocean (minimum 270 and 150 mg m$^{-2}$ yr$^{-1}$, respectively) (Barraqueta et al., 2019), the South Pacific Ocean (230 mg
m$^{-2}$ yr$^{-1}$) (Prospero, 1989), South Indian Ocean (240 mg m$^{-2}$ yr$^{-1}$) (Heimburger et al., 2012) and McMurdo Dry Valleys (490
mg m$^{-2}$ yr$^{-1}$) (Lancaster, 2002), while it is close to the estimated total dust deposition fluxes for the 63°S region (between
62°53'S and 63°53'S, along 170°6'W; 33 mg m$^{-2}$ yr$^{-1}$) (Measures and Vink, 2000). Accordingly, the other crustal elements (P,
Ti, V, Mn, Ce) were proportional to the dust dry deposition flux and showed extremely low values.

To examine the potential importance of atmospheric dust input to the particulate elemental concentrations in surface waters
of the West Antarctic Peninsula shelf region, the maximum possible suspended particulate Al concentrations in surface
seawater contributed by dust were estimated. We used the maximum Al dry deposition flux of 2300 µg m$^{-2}$ yr$^{-1}$ (Table 3),
assumed that dry deposition accounted for 40% of total deposition (Gao et al., 2003) and assumed no settling loss from the
mixed layer of mean depth 24 m (Eveleth et al., 2017) over a 4-month summer season. By this calculation, the accumulated
concentration of suspended particulate Al contributed by total atmospheric deposition over this period is 3 nmol kg$^{-1}$. While
suspended Al reaches 135 nmol kg$^{-1}$ in coastal surface waters close to the peninsula, outer shelf and off-shelf surface waters
generally have concentrations <5 nmol kg$^{-1}$ (Annett et al., 2017), suggesting that this upper limit estimated dust flux could
account for a substantial portion of observed surface ocean lithogenic particle concentration. While the mean Al flux is just
20% of this maximum, and settling loss from the mixed layer over the course of the summer may be significant, it is also
possible that the residence time of particulate Al in the surface layer exceeds the summer season, by analogy to the North
Atlantic (Jickells et al., 1999), allowing greater accumulation and concentrations than our simple calculation indicates. This
rough estimate suggests that atmospheric dust deposition, possibly dominated by regional sources on the Antarctic continent
itself, may contribute a significant fraction of suspended particulate concentrations of lithogenic elements in the surface waters
of the outer shelf and the proximal pelagic Southern Ocean.

## 4 Conclusions and Implications

Results from this study indicate that trace elements in aerosols over Palmer Station during the austral summer were primarily derived from (1) the regional crustal sources, which includes P-enriched soil resuspension, (2) remote anthropogenic emissions in South America , and (3) seawater. Remote crustal sources and local contaminated soil may also contribute to aerosol trace elements at Palmer Station. The particle size distributions of crustal elements, including Al, P, Ti, V, Mn, and Ce, were all concentrated in coarse-mode, suggesting strong regional emissions likely from ice-free areas on the Antarctic Peninsula and its associated islands. In some samples, Al, V, and Mn also had a secondary peak in the fine-mode fraction, likely derived from a distinct source through long-range transport. We speculate that the modest enrichment of aerosol P over its crustal ratio to Al was caused by the resuspension of regional soils that are P-enriched as a result of the impact of nearby bird colonies. The elements Ca, $Na^+$, and $K^+$, had a single coarse-mode distribution, likely derived from sea-salt. Conversely, the size distribution of Pb occurred primarily in fine-mode. The airmass back trajectories show that samples with high concentrations of Ni, Cu, Zn and Pb were influenced by air masses passing through southern South America and the South Pacific Ocean. The total dust deposition flux ($\sim$10 mg m$^{-2}$ yr$^{-1}$) during austral summer, estimated from the Al concentrations obtained in this study and an assumption of relative wet deposition, suggests that dust deposition plays a minor role in the concentrations of trace elements in coastal seawater around the West Antarctic Peninsula but may be more important in offshore regions. As the role of wet deposition is unquantified at present and remains poorly constrained for this region, the total deposition fluxes of trace elements during the austral summer could exceed the dry deposition fluxes reported here. Therefore, the importance of atmospheric deposition of trace elements to coastal West Antarctic Peninsula may need to be re-evaluated with additional observations of wet deposition. The Antarctic Peninsula is experiencing rapid climate change, with the expansion of ice-free areas and more frequent shipboard tourism with unknown impacts on aerosol chemistry in this region. To quantify future changes in in the atmospheric position and the impact in this region, long-term atmospheric observations of aerosol chemical and physical properties along with coupled studies of the ocean-atmosphere interactions would be needed.

*Data availability.* The data used in this paper has been submitted to the U.S. Antarctic Program Data Center (https://doi.org/10.15784/601370).

*Author contributions.* YG conceived the research. YG and SY prepared field sampling and collected aerosol samples. SF and YG digested samples. KB and RMS conducted sample analyses. SF wrote the 1st draft of the manuscript. YG and RMS edited the drafts. All authors contributed to writing and approved the submission.

*Competing interests.* The authors declare that they have no conflict of interest.

*Acknowledgements.* We thank Hugh Ducklow for encouragement of this research, Rafael Jusino-Atresino, Pami Mukherjee and

Guojie Xu for participating in fieldwork preparation, and Jianqiong Zhan for assisting with meteorological data analyses. This

work would not have become possible without the dedication and support from the staff of Palmer Station and the US Antarctic

Program.

*Financial support.* The study was supported by the US National Science Foundation Grant OPP-1341494 to YG.

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

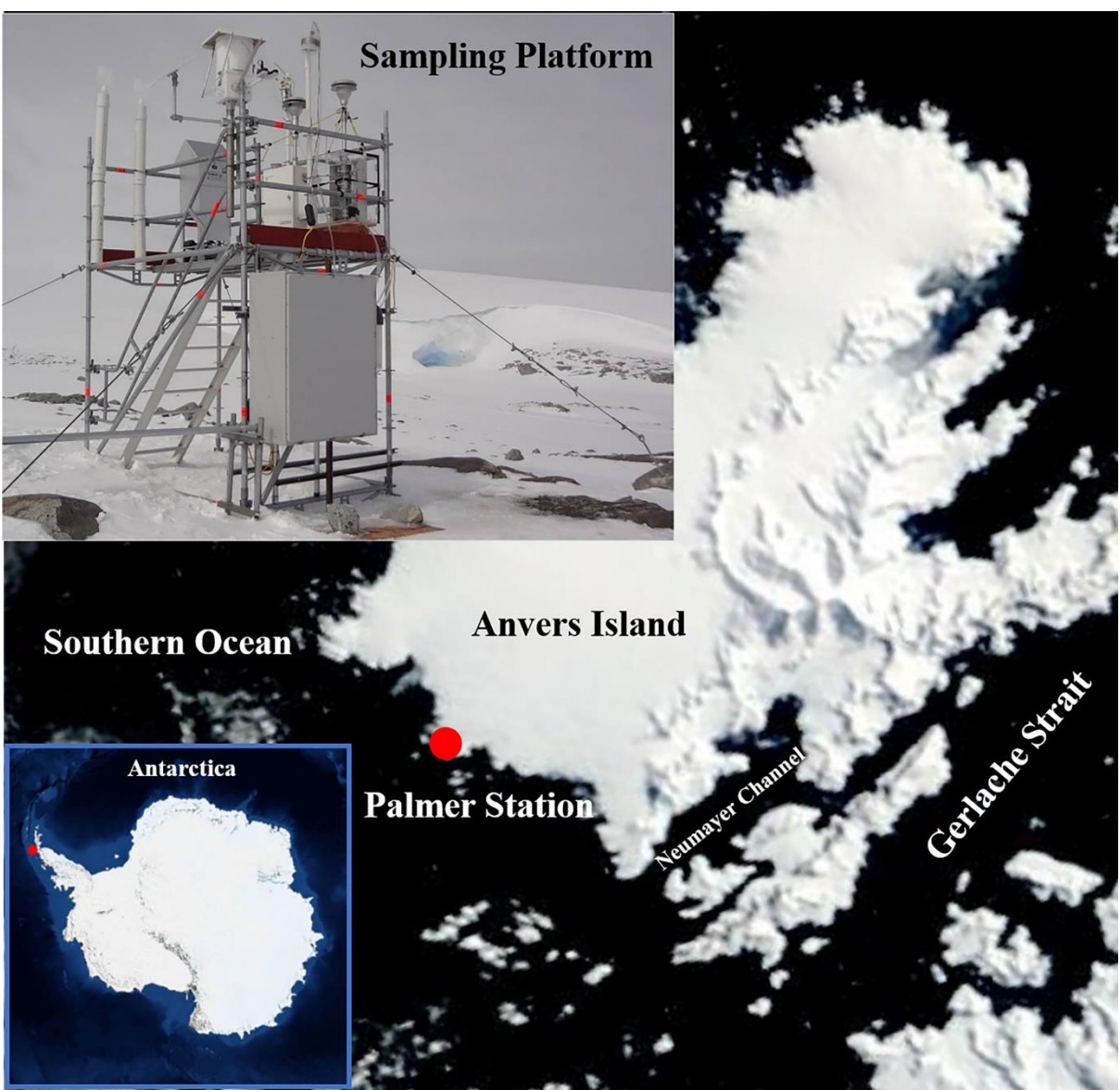

**Figure 1: Sampling site at Palmer Station (red dot) with inset photograph of sampling platform (Gao et al., 2020) (satellite image credits: NASA).**


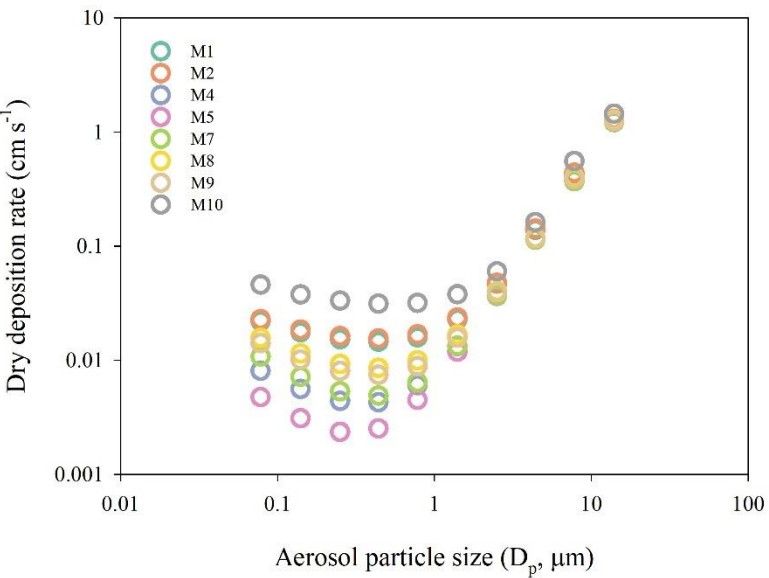

**Figure 2: Median dry deposition rates of each size class of aerosol particles for samples M1-M10, collected between Nov 2016 and Jan 2017. The dry deposition rate is a function of aerosol particle size (Dp), but also depends on meteorological conditions (see text).**

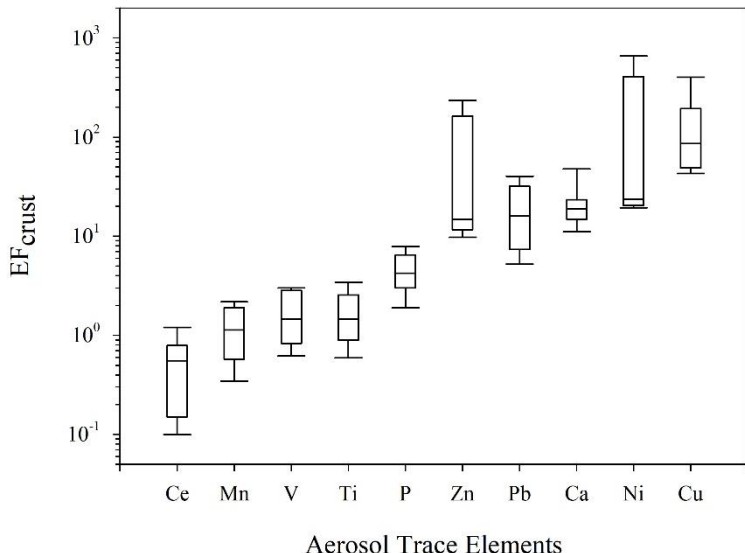


**Figure 3: Box plots of EF$_{crust}$ for trace elements in aerosols collected at Palmer Station between Nov 2016 and Jan 2017. The central horizontal line is the mean value, and the bottom and the top of each box are the 25th and 75th percentiles. The upper and lower horizontal bars indicate the 5th and 95th percentiles of the data.**

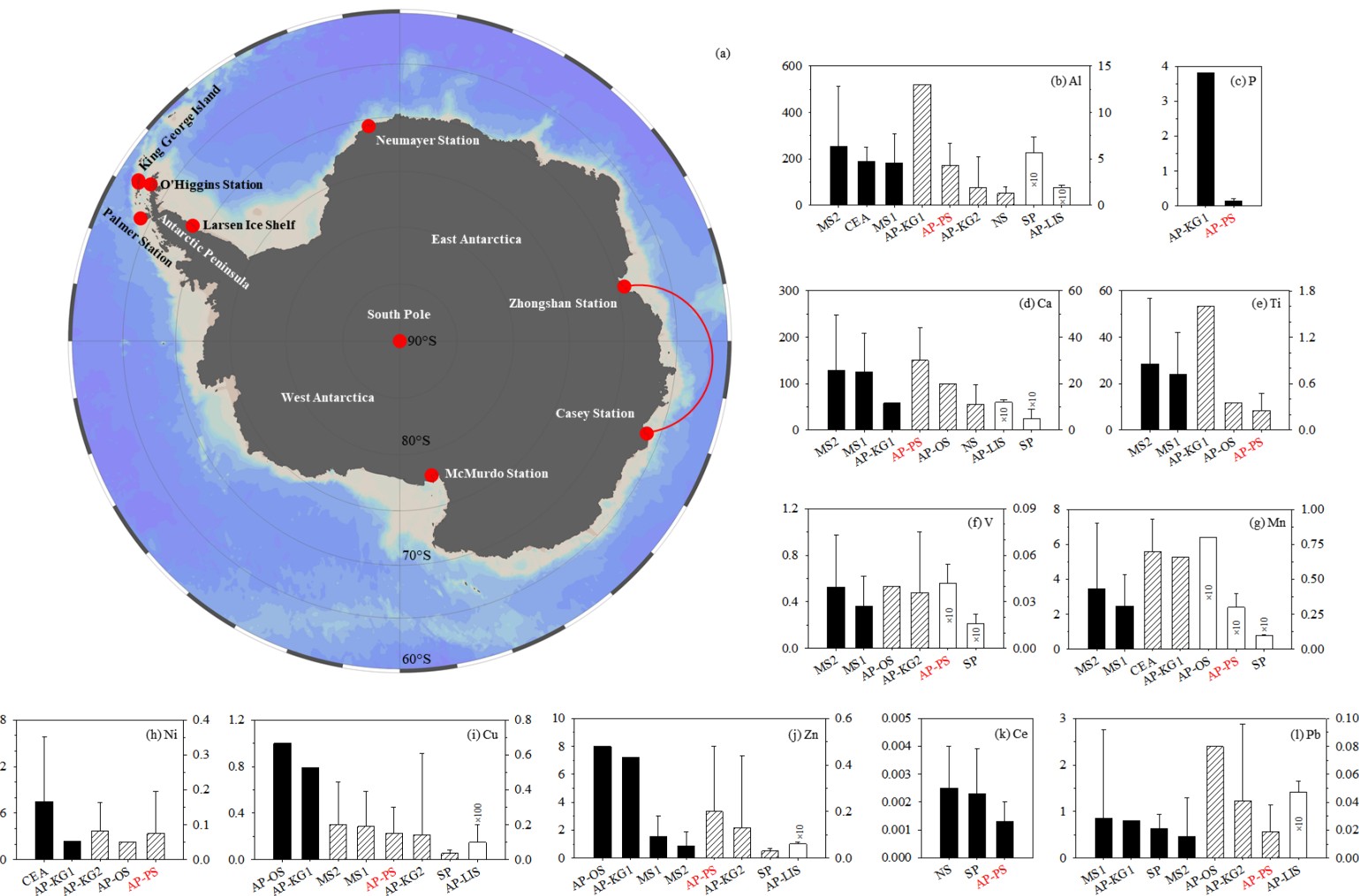

**Figure 4: Comparison of the concentrations of aerosol trace elements (ng m$^{-3}$) over Antarctica at coastal land-based sites. The cruise from Zhongshan Station to Casey Station was used to represent coastal East Antarctica. All concentrations are sorted in descending order from left to right. The left y-axis shows concentrations for the black bars, whereas the right y-axis corresponds to the striped and white bars. Some extremely low values are multiplied by 10 or 100 for display purposes, as marked above the corresponding white bars. Error bars show the standard derivation of the trace element concentrations in each study if available. Data are from**
**observations conducted at South Pole (SP) (Zoller et al., 1974), Hut Point site (MS1, PM$_{10}$) and Radar Sat Dome site (MS2, PM$_{10}$) at McMurdo Station (Mazzera et al., 2001), on a cruise between Zhongshan Station and Casey Station in the Coastal East Antarctica (CEA) (Xu and Gao, 2014), at Neumayer Station (NS) (Weller et al., 2008), and at 5 sites on the Antarctic Peninsula (AP), including Comandante Ferraz Antarctic Station (KG1) (Artaxo et al., 1990; Artaxo et al., 1992) and King Sejong Station (KG2) (Mishra et al., 2004) at King George Island, O'Higgins Station (OS, PM$_{2.5}$) (Préndez et al., 2009), Larsen Ice Shelf (LIS) (Dick, 1991), and Palmer**
**Station (PS, this study, marked by red arrow) at Anvers Island. Artaxo et al. (1990); Artaxo et al. (1992); Mazzera et al. (2001), and**

**Préndez et al. (2009) used X-ray fluorescence to measure the total concentrations of trace elements. All the other studies applied acid digestion methods. Our study at Palmer Station is marked in red.**

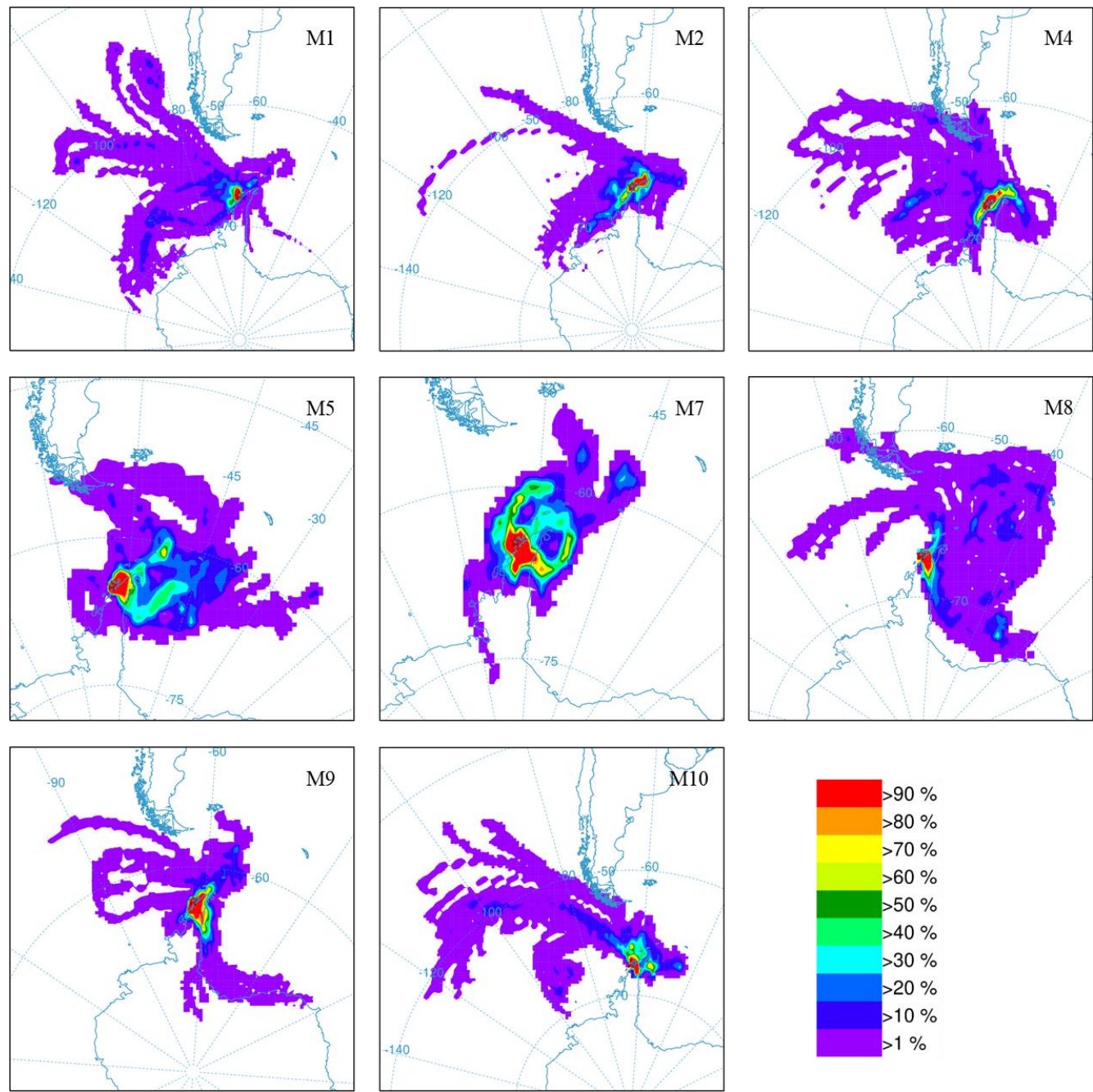


**Figure 5: Frequencies of 72-hour air mass back trajectories for samples collected at Palmer Station: M1 (Nov 19 – Nov 26, 2016), M2 (Nov 26 – Dec 4, 2016), M4 (Dec 09 – Dec 17, 2016), M5 (Dec 17 – Dec 24, 2016), M7 (Jan 1 – Jan 8, 2017), M8 (Jan 8 – Jan 15, 2017), M9 (Jan 15 – Jan 23, 2017), and M10 (Jan 23 – Jan 30, 2017).**

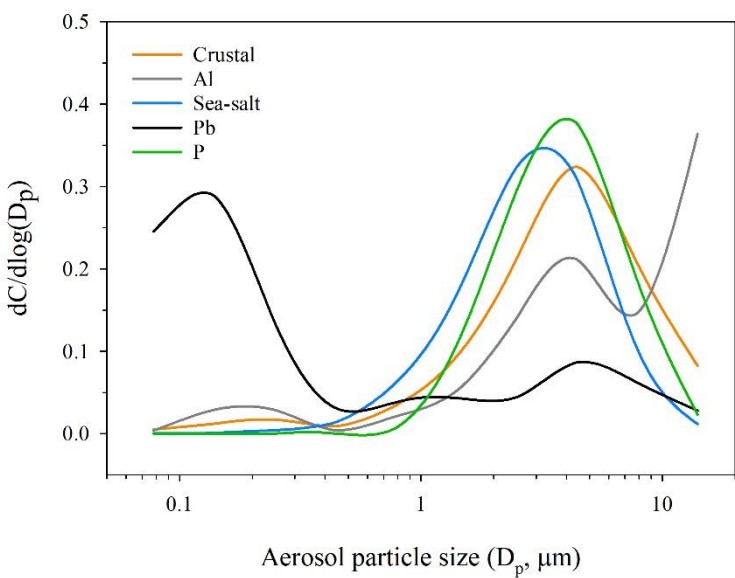


**Figure 6. The results of k-means clustering on the normalized mean aerosol particle size distributions. The crustal cluster includes Ti, V, Mn, Ce and the sea-salt cluster includes Na$^+$, K$^+$, Ca.**




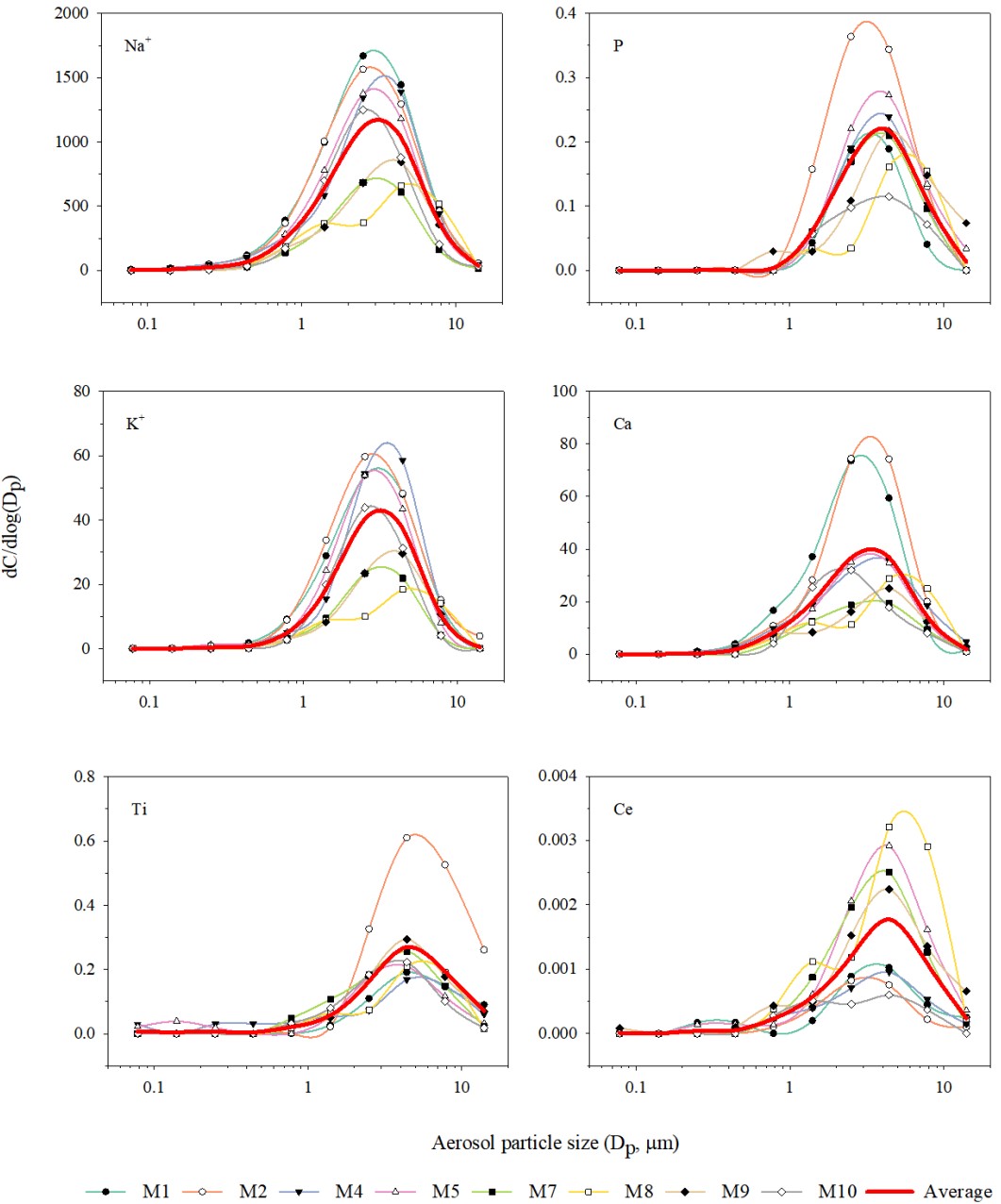

**Figure 7: Particle size distributions of the trace elements showing single-mode distribution in aerosols during Nov 2016 – Jan 2017 austral summer over Palmer Station. dC/dlog($D_p$) is the normalized concentration.**

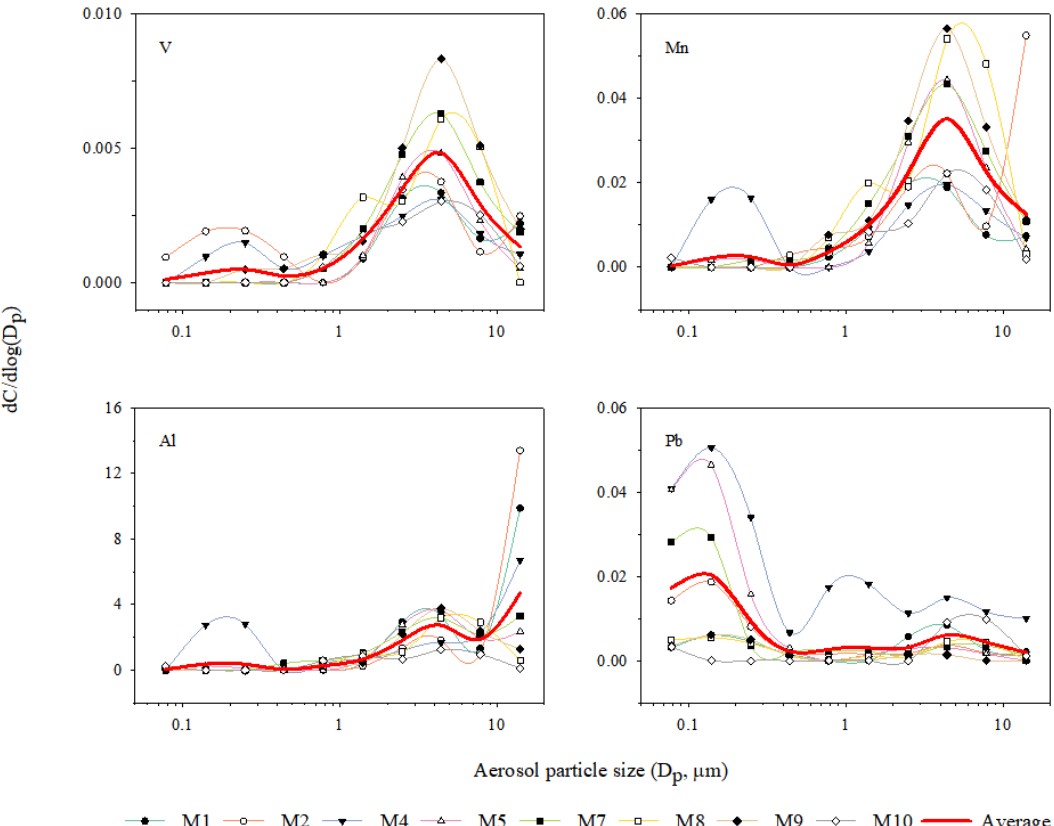

**Figure 8: Particle size distributions of the trace elements showing bimodal and trimodal distribution in aerosols during Nov 2016 – Jan 2017 austral summer over Palmer Station. dC/dlog(D$_P$) is the normalized concentration.**

**Table 1. Sampling periods and meteorological conditions for individual samples (Gao et al. 2020).**

| Sample ID | Sampling Period | %Actual Sampling Time* (%) | Sampling Volume (m³) | Wind Speed (m s⁻¹) | Air Temperature (°C) | Relative Humidity (%) | Air Pressure (hPa) | Precip. (mm d⁻¹) | Solar Intensity (W m⁻²) |
|---|---|---|---|---|---|---|---|---|---|
| M1 | 11/19/2016-11/26/2016 | 78 | 225 | 5.7 (0.7-22) | 0.1 (-2.2-4.3) | 87 (52-100) | 990 (976-1011) | 4.19 | 166 (0-675) |
| M2 | 11/26/2016-12/04/2016 | 71 | 233 | 4.9 (0.4-21) | 0.7 (-2.2-5.8) | 82 (45-100) | 982 (959-997) | 1.37 | 191 (0-847) |
| M4 | 12/09/2016-12/17/2016 | 80 | 266 | 3.5 (0.1-15) | 0.7 (-3.3-5.6) | 83 (53-100) | 985 (969-998) | 0.37 | 207 (1-854) |
| M5 | 12/17/2016-12/24/2016 | 88 | 264 | 2.5 (0.3-9.8) | 2.0 (-0.7-5.5) | 77 (49-100) | 988 (972-996) | 0.05 | 195 (1-723) |
| M7 | 01/01/2017-01/08/2017 | 85 | 244 | 4.1 (0.2-13) | 1.8 (-0.7-3.8) | 72 (49-99) | 987 (972-997) | 0.09 | 272 (0-857) |
| M8 | 01/08/2017-01/15/2017 | 94 | 283 | 5.4 (0.7-16) | 2.6 (0.3-5.5) | 64 (46-95) | 982 (973-991) | 0 | 225 (0-792) |
| M9 | 01/15/2017-01/23/2017 | 87 | 285 | 5.0 (0.2-15) | 2.1 (-1.0-5.3) | 65 (42-86) | 987 (981-997) | 0.15 | 260 (0-829) |
| M10 | 01/23/2017-01/30/2017 | 98 | 279 | 8.9 (0.7-21) | 4.1 (1.7-7.1) | 79 (65-90) | 981 (969-997) | 3.7 | 110 (0-594) |

*Actual sampling time / total sampling time × 100%

**Table 2. Concentrations of trace elements and ions in aerosols over Palmer Station, West Antarctic Peninsula.**

| Element/Ion | Units | Size Fraction | M1 | M2 | M4 | M5 | M7 | M8 | M9 | M10 | Mean | S.D |
|---|---|---|---|---|---|---|---|---|---|---|---|---|
| Al | ng m$^{-3}$ | Coarse | 6.9 | 7.9 | 4.4 | 3.3 | 3.7 | 2.2 | 2.5 | 0.83 | 4.0 | 2.4 |
| | | Fine | ND | ND | 1.6 | 0.13 | 0.30 | 0.11 | 0.17 | 0.41 | 0.34 | 0.53 |
| | | Sum | 6.9 | 7.9 | 6.0 | 3.4 | 4.0 | 2.3 | 2.7 | 1.2 | 4.3 | 2.4 |
| P | pg m$^{-3}$ | Coarse | 110 | 250 | 150 | 180 | 130 | 95 | 140 | 85 | 140 | 53 |
| | | Fine | ND | ND | ND | ND | ND | ND | 15 | ND | 1.9 | 5.3 |
| | | Sum | 110 | 250 | 150 | 180 | 130 | 95 | 160 | 85 | 150 | 54 |
| Ca | ng m$^{-3}$ | Coarse | 42 | 47 | 25 | 24 | 14 | 18 | 15 | 20 | 26 | 12 |
| | | Fine | 8.9 | 5.4 | 5.4 | 3.2 | 2.4 | 3.1 | 3.9 | 2.0 | 4.3 | 2.2 |
| | | Sum | 50 | 53 | 31 | 27 | 16 | 21 | 19 | 22 | 30 | 14 |
| Ti | pg m$^{-3}$ | Coarse | 150 | 800 | 120 | 150 | 170 | 130 | 190 | 94 | 230 | 230 |
| | | Fine | ND | ND | 48 | 20 | 24 | 9.0 | 16 | 65 | 23 | 23 |
| | | Sum | 150 | 800 | 170 | 170 | 190 | 140 | 210 | 160 | 250 | 220 |
| V | pg m$^{-3}$ | Coarse | 3.2 | 3.6 | 2.4 | 3.1 | 4.8 | 4.1 | 5.3 | 2.4 | 3.6 | 1.1 |
| | | Fine | ND | 1.4 | 1.2 | ND | 0.27 | 0.53 | 0.77 | 0.28 | 0.57 | 0.55 |
| | | Sum | 3.2 | 5.0 | 3.7 | 3.1 | 5.0 | 4.6 | 6.1 | 2.7 | 4.2 | 1.2 |
| Mn | pg m$^{-3}$ | Coarse | 17 | 42 | 15 | 27 | 32 | 34 | 35 | 14 | 27 | 10 |
| | | Fine | 1.2 | 2.3 | 8.2 | 0.90 | 2.9 | 3.6 | 3.9 | 2.8 | 3.2 | 2.3 |
| | | Sum | 18 | 44 | 23 | 27 | 35 | 38 | 39 | 17 | 30 | 10 |
| Ni | pg m$^{-3}$ | Coarse | ND | 320 | ND | 17 | 23 | ND | ND | ND | 45 | 110 |
| | | Fine | 37 | ND | ND | ND | ND | ND | ND | 200 | 30 | 70 |
| | | Sum | 37 | 320 | ND | 17 | 23 | ND | ND | 200 | 75 | 120 |
| Cu | pg m$^{-3}$ | Coarse | 170 | 200 | 17 | ND | ND | ND | ND | 120 | 63 | 86 |
| | | Fine | 69 | 280 | 92 | 52 | 110 | ND | 36 | 34 | 84 | 86 |
| | | Sum | 240 | 480 | 110 | 52 | 110 | ND | 36 | 150 | 150 | 150 |
| Zn | pg m$^{-3}$ | Coarse | 40 | 68 | 32 | 590 | 280 | 25 | ND | 46 | 140 | 200 |
| | | Fine | 51 | ND | 30 | 120 | 300 | ND | ND | 56 | 70 | 100 |
| | | Sum | 90 | 68 | 61 | 710 | 580 | 25 | ND | 100 | 200 | 280 |
| Ce | pg m$^{-3}$ | Coarse | 0.73 | 0.57 | 0.62 | 1.8 | 1.6 | 2.1 | 1.5 | 0.41 | 1.2 | 0.65 |
| | | Fine | 0.084 | 0.051 | 0.12 | 0.13 | 0.12 | 0.19 | 0.26 | 0.14 | 0.14 | 0.064 |
| | | Sum | 0.81 | 0.62 | 0.74 | 2.0 | 1.7 | 2.2 | 1.8 | 0.55 | 1.3 | 0.69 |

| | | | | | | | | | | | | |
|---|---|---|---|---|---|---|---|---|---|---|---|---|
| Pb | pg m$^{-3}$ | Coarse | 5.1 | 2.4 | 13.8 | 2.2 | 2.4 | 2.7 | 0.70 | 5.0 | 4.3 | 4.1 |
| | | Fine | 3.8 | 12 | 46 | 28 | 17 | 4.5 | 4.3 | 1.7 | 15 | 16 |
| | | Sum | 9.0 | 14 | 60 | 30 | 19 | 7.2 | 5.0 | 6.6 | 19 | 19 |
| Na$^+$ | ng m$^{-3}$ | Coarse | 1100 | 1000 | 900 | 870 | 430 | 450 | 520 | 730 | 749 | 260 |
| | | Fine | 220 | 210 | 160 | 120 | 75 | 99 | 89 | 90 | 130 | 57 |
| | | Sum | 1300 | 1200 | 1100 | 1000 | 510 | 550 | 610 | 820 | 890 | 310 |
| K$^+$ | ng m$^{-3}$ | Coarse | 34 | 39 | 34 | 31 | 14 | 12 | 17 | 24 | 26 | 10 |
| | | Fine | 4.9 | 4.5 | 2.5 | 3.2 | 1.6 | 1.8 | 1.5 | 1.4 | 2.3 | 1.1 |
| | | Sum | 39 | 43 | 37 | 34 | 15 | 14 | 19 | 25 | 28 | 11 |


**Table 3. Estimates of dry deposition velocities and dry deposition fluxes of trace elements in aerosols over Palmer Station, West Antarctic Peninsula.**

|  | Element | Velocity (cm s$^{-1}$) | | Flux ($\mu$g m$^{-2}$ yr$^{-1}$) | |
|---|---|---|---|---|---|
|  |  | Range | Mean | Range | Mean |
|  | Dust |  |  | 650-28000 | 5500$\pm$5100 |
| EF<10 | Al | 0.14-1.6 | 0.49 | 52-2300 | 440$\pm$410 |
|  | P | 0.092-0.52 | 0.17 | 3.3-33 | 7.5$\pm$4.4 |
|  | Ti | 0.14-0.92 | 0.27 | 7.1-120 | 18$\pm$15 |
|  | V | 0.14-0.85 | 0.26 | 0.16-0.77 | 0.32$\pm$0.11 |
|  | Mn | 0.18-1.1 | 0.28 | 0.94-11 | 2.5$\pm$1.6 |
|  | Ce | 0.10-0.65 | 0.20 | 0.017-0.31 | 0.084$\pm$0.048 |
| EF>10 | Ca | 0.073-0.41 | 0.13 | 580-6600 | 1200$\pm$660 |
|  | Pb | 0.0097-0.83 | 0.11 | 0.018-1.5 | 0.30$\pm$0.23 |
|  | Ni* |  | 0.11-0.49 | 3.9-17 |  |
|  | Cu* |  | 0.11-0.49 | 6.0-25 |  |
|  | Zn* |  | 0.11-0.49 | 8.1-36 |  |

*The range of dry deposition fluxes of Ni, Cu, and Zn was estimated based on Pb and Al dry deposition velocities, discussed in the text.