# Peer review of "Concentrations, Particle-Size Distributions, and Dry Deposition Fluxes of Aerosol Trace Elements over the Antarctic Peninsula in Austral Summer"

_Atmospheric Chemistry and Physics, 2020_

## Referee Comment (RC1) · Holly Winton (Referee) · 21 Sep 2020

Fan et al. report size segregated aerosol concentrations and dry deposition fluxes for a suite of trace elements and cations from a land-based station on the Antarctic Peninsula over the 2016-2017 austral summer. Aerosol iron solubility data from these samples has been previously published by Gao et al. (2020). While the timeseries is short (8 sets of size segregated samples each with a sampling duration of 1 week), the study fills a gap in data availability of aerosol trace element levels in this sector of Antarctica and also in our understanding of the deposition pathways. The authors have estimated dry deposition velocities for each trace element at the sampling site,

providing a more reliable estimate of dry deposition fluxes than previous studies which typically use a generic value not necessarily appropriate for the region with a large uncertainty. The study also reports aerosol size distribution of the trace elements which is extremely valuable data, also scarce in this region, providing important insights into the source and transport pathways of aerosols in pristine airmasses. The manuscript is well written, has a logical structure and provides important data to a wide-ranging community of researchers from atmospheric chemists, biogeochemists and ice core scientists. I recommend publication in ACP after a few suggestions for improvement have been addressed. I hope the suggestions below will strengthen the manuscript.

Suggestions for improvement Introduction: It would be helpful to add a short section on what is known about aerosol removal processes in the Southern Ocean and where the gaps are in our understanding. I also think it would be worth briefly summarizing the aerosol iron data here e.g. mineral dust source and probable source region. Discussion: The discussion on aerosol sources and, in particular, particle size distribution needs clarification and further investigation. The particle size distribution is a great dataset but at the moment it hasn't been fully utilized. Incomplete referencing and a limited explanation of the particle distribution let the discussion down. The authors could improve this section with a statistical cluster analysis of the physical characteristics of particle size distributions and/or a thorough understanding/comparison of the Antarctic and Southern Ocean literate on this topic. Currently, there is no discussion on what the size distribution us about the atmospheric transport mechanisms of the individual trace elements. Also, the conclusions don't come through strongly enough in the discussion. Conclusions and implications: It's fantastic to see the size distribution data. Do you have plans to continue aerosol monitoring at this land-based sampling station to make a long-term record of size segregated trace elements? In this section, you could recommendation future aerosol studies investigate the size distribution to provide additional information on the source and atmospheric transport. What are the implications for the underestimation and overestimation of dry deposition velocities in previous aerosol flux estimates and climate modelling for the region? What about the

implications of your new particle size distribution data for new particle formation studies? Throughout: Please make it clear throughout the study that you are reporting total acid digestible trace element concentrations and water soluble cations. Please report Ca and K as ions Ca2+ and K+ rather than elements throughout the manuscript.

Technical corrections L11 Suggest replacing with "local mineral dust, long-range transported aerosol and sea salt aerosol." What about local P biogenic emissions? L26 Define course mode. L32 Include sea salt as a source of aerosol. L37 A number of local dust sources in Antarctica are increasingly being quantified. See Delmonte et al. (2019) for a recent review. Delmonte, B., Winton, H., Baroni, M., Baccolo, G., Hansson, M., Andersson, P., Baroni, C., Salvatore, M.C., Lanci, L. and Maggi, V., 2020. Holocene dust in East Antarctica: Provenance and variability in time and space. The Holocene, 30(4), pp.546-558. L47 Include a refence to McMurdo Sound here as the dustiest region in Antarctica. L62 "...new sources of aerosol trace elements by warming which exposes a greater area of ice-free land and by..." L68 Mention that a detailed description of the aerosol sampling and protocols to mitigate contamination in the pristine environment can be found in Gao et al. (2020). L100 Please report precision here or in Table 2. L107 Did you measure other cations and anions on these samples? If you did, will these data be included in your US Antarctic Program Data Centre data publication? I understand they may not be relevant for this manuscript but I'm sure they would be beneficial to other researchers if the data is available. Why did you not measure Ca an K by ICP-MS? L121 Why did you select Al over Ti as the crustal reference? L128 "...was calculated from the concentration of the trace element in the air and dry..." L141 What are the environmental factors? L142 Please report the uncertainty of your dry deposition velocity estimates. Also, where can the reader find the dry deposition velocity values for each element? L160-185 Can you use refences for P and V sources from the Southern Hemisphere? I'm not aware of any evidence that agricultural and industrial emissions from China are transported to Antarctica. It seems that P at your site is dominantly sourced from local Antarctic emissions rather than long-range transported agricultural or volcanic emissions. Do airmasses pass

over Antarctic soils, or seabird colonies? While ship emissions of V have been detected in aerosols in the Arctic and Atlantic where shipping is more frequent, is there any evidence of this in the Southern Ocean? While it's important to identify all possible sources of these elements, this section could be tightened up to clarify the most likely sources of this group of elements and rule out the unlikely ones to Antarctica. L186 and 242 Many of these elements occur naturally in the crust but in your samples they have a non-crustal source. I wonder if you could rename the headings to avoid confusion e.g. rename the group of elements into one of the three categories described in the abstract - local mineral dust, long-range transported aerosol and sea salt aerosol. L195 Again, please use refences for the Southern Ocean rather than Northern Hemisphere. Some suggestions for Pb: McConnell, J.R., Maselli, O.J., Sigl, M., Vallelonga, P., Neumann, T., Anschütz, H., Bales, R.C., Curran, M.A., Das, S.B., Edwards, R. and Kipfstuhl, S., 2014. Antarctic-wide array of high-resolution ice core records reveals pervasive lead pollution began in 1889 and persists today. Scientific Reports, 4(1), pp.1-5. Bollhöfer, A.F., Rosman, K.J.R., Dick, A.L., Chisholm, W., Burton, G.R., Loss, R.D. and Zahorowski, W., 2005. Concentration, isotopic composition, and sources of lead in Southern Ocean air during 1999/2000, measured at the Cape Grim Baseline Air Pollution Station, Tasmania. Geochimica et Cosmochimica Acta, 69(20), pp.4747-4757. L200-201 This sentence isn't clear whether you are suggesting that Ca is derived from seawater or another source? For the modern day, many coastal ice core sites suggest a marine source of Ca, where during glacial periods Ca is dominated by a long-range transported mineral dust. L208-211 Refer the reader to Fig. 4. L215-216 Al concentrations have been measured in local dust in snow on sea ice in McMurdo Sound which is the dustiest location in Antarctica. L232-235 Comparison to trace element concentrations in regions outside of the Southern Ocean or the Southern Hemisphere seems irrelevant. L235-237 These pieces of evidence further confirm your EF interpretation that P was mainly locally derived from soil and bird colonies proximal to the station. How did you calculate nss-K? Please state the limitations of using nss-K as a biomass burning tracer in the Southern Ocean as opposed to sites proximal to biomass/fires.

See other studies in the Southern Ocean e.g. nss-K at Cape Grim. Is there a better tracer of biomass burning that you have access to over the sampling period? Winton, V.H.L., Bowie, A.R., Edwards, R., Keywood, M., Townsend, A.T., van der Merwe, P. and Bollhöfer, A., 2015. Fractional iron solubility of atmospheric iron inputs to the Southern Ocean. Marine Chemistry, 177, pp.20-32. L243-255 Open paragraph with a sentence to let the reader know you are discussing Ni, Zn, Pb and Cu first. It's not clear what caused the large variability in this group of elements (Ni, Zn, Pb and Cu). Can you link these elements with air mass trajectories as you did with Pb? What anthropogenic activities emit Ni, Zn, Pb and Cu in South America? L258 Please compare to other studies of aerosol Pb in the Southern Ocean. L2643-264 "The low concentration of Pb observed in samples associated with trajectories that did not pass over Southern South America suggest that. . ." L265 ". . .long-range transported aerosol from Southern Hemispheric continental regions containing a mixture of anthropogenic and crustal emissions such a South America." L272 Do you mean K wasn't influenced by biomass burning. This should be mentioned before you use nss-K as a biomass burning tracer. L266-273 See modern day, coastal snow chemistry studies of Ca, Na and K derived from sea salt and sea ice to further back up the high abundance of marine derived elements at your site. Also, open the paragraph with a sentence to let the reader know you are discussing Ca, Na and K. End with a clear sentence stating the source(s). L277 Consider renaming "elements from the continents" to "mineral dust" L2278 First mention of combustion. You previously ruled out biomass burning. Please state which elements are combustion products in the introduction or discussion section on sources. L281 This group of elements all seem to have a coarse mode around the same size. Please include. L281-282 Is this reference for long-range transported dust? L280-289 This section could be considerably improved. Some elements have a secondary mode in some samples but a single mode in other samples e.g. Al, V, Mn, Pb. Please discuss the episodic deposition from multiple sources and include in your discussion that these elements can have more than one source in a particular airmass. Please plot size distributions for all elements in Figure 6. I can't see Na and K. Include in your discussion

a comparison with the literature to help attribute individual modes to sources. Some, but not all, references for Antarctic aerosol size distributions are below. I think the elements should be grouped differently according to the particle size distribution rather than the crustal vs non-crustal EF analysis. For example, crustal sources: P, Ti, Ca, and Ce have a single coarse mode distribution around XYZ um indicating. V and Mn have a bimodal distribution with a primary mode around XYZ um and secondary mode around ZXY um. Pb has a bimodal and sometimes trimodal distribution with a fine primary mode that is much finer than and mode in the other elements. Al, again, has a different distribution with a bimodal and sometimes trimodal distribution with mode similar to P, Ti, Ca, and Ce and an additional coarse model not seen in other elements. Also, Al exhibits a fine mode, similar to Mn and V, but only in one sample. Can you rule out local dust contamination for coarse Al concentrations? Could you compare the Al particle size distribution to mineral dust size distributions in Antarctic snow? A very interesting section could be developed about this comparison. Lachlan-Cope, T., Beddows, D.C., Brough, N., Jones, A.E., Harrison, R.M., Lupi, A., Jun Yoon, Y., Virkkula, A. and Dall'Osto, M., 2020. On the annual variability of Antarctic aerosol size distributions at Halley Research Station. Delmonte, B., Winton, H., Baroni, M., Baccolo, G., Hansson, M., Andersson, P., Baroni, C., Salvatore, M.C., Lanci, L. and Maggi, V., 2020. Holocene dust in East Antarctica: Provenance and variability in time and space. The Holocene, 30(4), pp.546-558. Fattori, I., Becagli, S., Bellandi, S., Castellano, E., Innocenti, M., Mannini, A., Severi, M., Vitale, V. and Udisti, R., 2005. Chemical composition and physical features of summer aerosol at Terra Nova Bay and Dome C, Antarctica. Journal of Environmental Monitoring, 7(12), pp.1265-1274. Virkkula, A., Teinilä, K., Hillamo, R., Kerminen, V.M., Saarikoski, S., Aurela, M., Koponen, I.K. and Kulmala, M., 2006. Chemical size distributions of boundary layer aerosol over the Atlantic Ocean and at an Antarctic site. Journal of Geophysical Research: Atmospheres, 111(D5). Fossum, K.N., Ovadnevaite, J., Ceburnis, D., Dall'Osto, M., Marullo, S., Bellacicco, M., Simó, R., Liu, D., Flynn, M., Zuend, A. and O'dowd, C., 2018. Summertime primary and secondary contributions to Southern Ocean cloud condensation nuclei. Scientific

reports, 8(1), pp.1-14. L318-319 Note that estimates of wet deposition fluxes are even scarcer than dry deposition fluxes! Please state the limitations with this assumption but note it is likely the best estimate we have. L321 Dome C is a high elevation site and receives fine dust with a mode around 2-3 um. L305-340 Please add errors on these fluxes. L323-327 Suggest removing the comparison to values outside of the Southern Ocean and replacing with a discussion on why the flux is at the lower end of the JRI estimate. L332 Add reference for the 40 % assumption. L331-333 What geographic area does the estimate represent? L337-341 Could you estimate the lower bound as well and report both upper and lower bounds for summer? This information would be useful to include in the abstract too. L343 This conclusion omits previous discussion of anthropogenic emissions. Please keep a consistent message throughout the manuscript. What about biogenic emissions of P here and in the abstract? L348-351 I don't see how you can make this claim without data prior to the commencement of shipping. There are regional differences in aerosol chemistry and dust fluxes around Antarctica. Perhaps rephrase indicating that future impacts of shipping and changes in the ice-free area have an unknown impact on aerosol chemistry in the region. Fig 1 and Table 1: These look like copies of Fig. 1 and Table 1 in Gao et al. (2020). Please state if the figure is reproduced from this publication. Fig. 2: State what the label stand for. Please provide more information in the caption. Fig. 3: Please state the time period the samples represent in the caption. Fig. 4: State that these are all land-based sampling sites except one cruise. Are these all total digestible concentrations? Why don't you add other data from cruises in the Southern Ocean? Fig 5: Why are you only showing air mass back trajectories of 4 samples? Please make the continent outline clearer as it is difficult to see under the air mass fetch regions. Fig. 6: Please provide more information in the caption. Group the elements into the size distribution patterns discussed in the text. State what the axis labels stand for and include a reference for the notation in the methods. Are you plotting the number or volume particle size distribution? Please plot size distribution for ALL elements. Tables 2-3: Could be moved to the supplement.

---

## Referee Comment (RC2) · Anonymous Referee #2 · 30 Sep 2020

This study investigated the concentrations, size distributions, and dry deposition fluxes of about 15 airborne trace elements in the Antarctic Peninsula. The results will be useful for the scientific community, especially considering the fact that the particulate-bound elemental analysis for remote areas is scarce. However, there are scopes of improving the data analysis approach that will make the conclusions of the study more reliable.

General comments:

1. The total sampling period of the study is rather short (∼2 months). Authors need to discuss the possible uncertainties/seasonal influence affecting the element concentrations, especially in the context of comparison with other studies.

2. The study used Al concentration (8% of the dust concentration) to calculate the dry and total deposition of dust in the region. A few major conclusions of the study were based on this assumption. The assumption could lead to uncertainties in the results as it is based on the Al concentration only and may not represent the crustal matter in the study region. Therefore, the authors should provide sufficient evidence to establish that this is a solid assumption. Mass reconstruction of soil/dust based on all available crustal elements concentration (rather than just the Al concentration) can also be considered.

3. Some figures and parts of the text should be revised to improve the clarity (e.g., Figures 4, Section 3.2.1, etc.). Specific line-by-line comments are as follows.

Specific comments:

Line 10: "The results show..." The reader would be wondering about the study methods/data analysis approach that leads to these results (i.e., enrichment calculation or statistical analysis/ source apportionment tool, etc.?).

Lines 11-13: "Elements dominated by a crustal source... reflecting the contributions of regional crustal sources" repetitive.

Line 13: The term "EFcrust" may not be familiar to all readers.

Line 13: "...coarse-mode particles (>1 $\mu$m)..." contradicts with lines 72-73 that states ''...those >=1.8 $\mu$m were summed to define coarse-mode particles...".

Lines 26-27: "It has been realized that the impact of coarse mineral dust has been underestimated..." I am not sure what argument is presented here.

Lines 54-56: "However,...are lacking." Is this the first study on elements sampled in Antarctica? If yes, state clearly, if not, briefly mention about the prior studies that sampled aerosol in this region and what this study introduces. The paragraph that

follows (lines 57-65) also does not give much idea about the knowledge gap that this study fills in.

Section 2, Line 66 onward: No mention of how the samples below "limits of detection (LOD)" were treated. Lines 77-78: "...wind direction inside the sector $\pm60°$ from the direction of the station and wind speed <2 m s-1." Unclear to me.

Line 91: "...Al, P, Ca, Ti, V, Mn, Ni, Cu, and Zn" this list looks different from the one mentioned in the abstract (line 9) that states "...Al, P, Ca, Ti, V, Mn, Ni, Cu, Zn, Ce, and Pb...".

Line 133: The summation is over what variable?

Line 135: So, Vd was calculated from the combination of two models?

Line 161: "The values of EFcrust for Ti, V, Mn, and Ce in aerosol samples were less than 10..." Why P is missing from this list where it has EFcrust<10 (Figure 3)?

Lines 170-171: "A similar phenomenon was observed at McMurdo Station where light-weight fuel oil was used that was not a significant source of V" not clear what was referred here.

Lines 173-185: My understanding is that the purpose of this paragraph is to establish that the enrichment of P is higher than other crustal elements. But based on the results (Figure 3), it appears that P enrichment is not significantly high in this study, compared to the other relevant studies.

Lines 197-198: "Hence, despite the recent increase in tourist ship traffic, it looks that Palmer Station was barely impacted by ship emissions" assuming this is correct, what is the reason for large variations of Ni (e.g., Figure 3)?

Lines 199-200: "Ca accounts for about 3.5% of the weight of Earth's crust, while Ca is also a conservative major ion in seawater" provide citation.

Line 215: "...the impact of the nearby McMurdo Dry Valleys" I could not understand

what impact was referred here.

Lines 217-218: "The concentrations of Ti and Mn ranged from…respectively." I am a little confused if the ranges are for individual elements or both elements combined.

Lines 220-223: "…but comparable to the concentrations… (Figure 4e and g)" is it applicable to both Ti and Mn? Figures 4e and 4g do not support this claim. Mn concentration at AP-OS is more than double of AP-PS (Figure 4g).

Line 232: "…the P values…" P concentrations?

Line 235-236: "Comparing global aerosol P concentrations…as those over the Central Pacific Ocean (Chen, 2004)" provide the concentration values from the referred study.

Lines 236-237: "Confirming that Palmer Station was little influenced by aerosols derived from biomass burning through long-range transport, the calculated non-sea-salt-K was indistinguishable from zero." I could not understand what this means.

Lines 239-241: "…suggesting that aerosol crustal elements observed at Palmer Station were impacted by sources in that region (Figure 5)." Why this argument applies to only crustal elements?

Lines 263-264: "The low concentrations of heavy metals observed during this study suggest that local anthropogenic emissions were negligible." Which metals you are referring to?

Lines 264-265: "Thus the major source of non-crustal elements in aerosols over the study region may be long-range transport from regions impacted by anthropogenic emissions" very weak conclusion as it is only based on Pb variation.

Line 271: "…27 and 26, in seawater" respectively?

Lines 272-273: "The results suggest that Ca was dominated by sea-salt aerosol…" what about K?

Line 276: "classified into three groups based on their potential dominant sources" is this classification is in the context of section 3.1? If yes, it should be stated clearly. If that's not the case, provide justification of the grouping.

Lines 301-302: "The mass distributions of sea-salt elements (Ca, Na and K) as the third group were dominated by coarse-mode particles with diameters 2.5–7.8 $\mu$m (Figure 6)" I could not find the size distributions of Na and K in Figure 6!

Line 303: "...the correlation between the total concentrations of Ca and Na was strongly positive (R2±0.82, p-value < 0.01)..." was it based on the 8 pairs of samples presented in Table 3? What about the correlations of other elements (such as K) with Na? Like Ca, If K is also associated with seasalt (as suggested in lines 270-272) one would expect a good K-Na correlation.

Lines 311-313: "The rough estimates of the dry deposition fluxes of Ni, Cu, and Zn at Palmer Station..." should mention few values from the literature so the readers get an idea of how large or small the values are.

Lines 313-315: "The estimated dry deposition fluxes of total continental dust... among the lowest globally (Lawrence and Neff, 2009)" should mention a few global values.

Lines 316-319: "...precipitation scavenging accounted for about 40~60% of the total deposition..." are these fractions yearly average? Contributions of wet deposition to the total flux is a strong function of season. This study is limited to only two months of sampling. Authors need to discuss the likely uncertainties involved with extrapolating the short-term dry or wet deposition flux to yearly contributions.

Lines 318-319: "Assuming this wet deposition fraction applies to the Antarctic Peninsula region, we approximate roughly a total dust flux of 10 mg m-2 yr-1" I understand that this is an estimation, but how was it obtained? Assuming dry/total as 0.4? or 0.6?

Lines 329-341: Authors need to carefully revise this entire paragraph to ensure it is readable.

Lines 346-348: "Most of the samples collected during this study were impacted by air masses originating around or passing over Northern Antarctic Peninsula..." This claim is not supported by any strong evidence. Airmass trajectories associated with 4 of the total samples were presented in Figure 5.

Lines 354-357: "As the role of wet deposition is unquantified at...may need to be re-evaluated." I could not understand what this sentence is referring to.

Figure 2: Missing proper x-axis label. In addition, the figure caption should mention what the legends (M1...M10) are referring to.

Figure 4: This figure should be revised. 1) What is AP-OS1 shown in Figure 4(g)? 2) What are the vertical lines representing? 3) The acronyms of the sites mentioned in the caption should be consistent with the ones shown on figures (e.g., AP-PS or PS). 4) As mentioned in the text (e.g., line 211), air mass samples from many of the previous studies used for the comparison correspond to PM10 or even PM2.5. The figures should clearly indicate this (e.g., AP-OS(PM2.5) ).

Figure 5: Why is this figure showing 4 samples only?

Table 4: Should include a few extra columns showing literature values.
* * *

---

## Author Comment (AC2) · 3 Dec 2020

Reviewed by anonymous reviewer

**Major Comments:**

1. The total sampling period of the study is rather short (2 months). Authors need to discuss the possible uncertainties/seasonal influence affecting the element concentrations, especially in the context of comparison with other studies.

> ***Response:*** This study focused on atmospheric trace elements during the austral summer in Antarctic Peninsula. Given that the snow/ice cover could limit local dust emissions, we expect lower concentrations of crustal elements in austral winter. However, it is difficult to evaluate how the other trace elements may vary seasonally without seasonal observations. In the text, we added the detailed study periods (e.g. summer mean concentration or yearly mean concentration) for previous studies of crustal element concentrations, for comparison with our results. In addition, we added "during the austral summer" in the title of this paper to emphases the study season.

2. The study used Al concentration (8% of the dust concentration) to calculate the dry and total deposition of dust in the region. A few major conclusions of the study were based on this assumption. The assumption could lead to uncertainties in the results as it is based on the Al concentration only and may not represent the crustal matter in the study region. Therefore, the authors should provide sufficient evidence to establish that this is a solid assumption. Mass reconstruction of soil/dust based on all available crustal elements concentration (rather than just the Al concentration) can also be considered.

> ***Response:*** The concentrations of Al have been used to represent the contribution from crustal emissions at many sites in Antarctica, such as Antarctic Peninsula (Dick, 1991), McMurdo (Lowenthal et al., 2000), and the South Pole (Zoller et al., 1974). We searched for previous studies that reported rock or soil composition in Antarctica and learned that West Antarctica has a complicated geological history and many different rock types (Pereira et al., 2018). Nelson (1966) reported the percentage of Al ranged from 7.83 % to 10.42 % in 16 rock samples collected at James Ross Island, Antarctic Peninsula. On average, Al accounted for $8.86 \pm 0.79$ % in the total mass of rock. This number is about 10% different from the average upper crustal abundance of Al, 8.04% (Taylor and McLennan, 1995). In addition, there is a portion of crustal material

contributed by long-range transport, for which we don't know the source region. In this case, we decided to use the average crustal abundance of Al to estimate the dry deposition flux. Such uncertainty is small compared with the uncertainty of dry deposition estimation (a factor of 2 or 3). Given the undetermined mineralogy of the dust we collected, we estimate that dust mass determined by our measurements of aerosol Al and its assumed stoichiometry, as the reviewer suggests, would be at least as uncertain as our approach. Therefore we did not revise our methodology in this case.

3. Some figures and parts of the text should be revised to improve the clarity (e.g., Figures 4, Section 3.2.1, etc.). Specific line-by-line comments are as follows.

> ***Response:*** We have revised Figure 4, highlighted our study site at Palmer Station in red to improve the clarity. We have also added more detailed information in the figure caption.

**Specific comments:**
Line 10: "The results show. . ." The reader would be wondering about the study methods/data analysis approach that leads to these results (i.e., enrichment calculation or statistical analysis/ source apportionment tool, etc.?).

> ***Response:*** We have added the crustal enrichment factor and k-means clustering here as the main approach that to get these results. Please see lines 11-12.

> ***Modified text:*** lines 11-12: "The crustal enrichment factors ($EF_{crust}$) and k-means clustering results of particle size distributions show…"

Lines 11-13: "Elements dominated by a crustal source. . . reflecting the contributions of regional crustal sources" repetitive.

> ***Response:*** We agree with the reviewer and have revised the part of the text. Please see lines 13-15.

> ***Modified text:*** lines 13-15: "Elements derived from crustal sources (Al, P, Ti, V, Mn, Ce) with $EF_{crust}<10$ were dominated by the coarse-mode particles (>1.8 µm) and peaked around 4.4 µm in diameter, reflecting the regional contributions."

Line 13: The term "EFcrust" may not be familiar to all readers.

> ***Response:*** We have added "crustal enrichment factor ($EF_{crust}$)" to define this term in line 11.

> ***Modified text:*** line 11: "The crustal enrichment factors ($EF_{crust}$) and…"

Line 13: ". . .coarse-mode particles (>1 µm). . ." contradicts with lines 72-73 that states

''. . .those >=1.8 µm were summed to define coarse-mode particles. . .''.

> ***Response:*** This typo has been corrected. Please see line 14.

> ***Modified text:*** line 14: "…the coarse-mode particles (>1.8 µm)…"

Lines 26-27: "It has been realized that the impact of coarse mineral dust has been underestimated. . ." I am not sure what argument is presented here.

> ***Response:*** We have revised this sentence. Please see lines 28-29.

> ***Modified text:*** lines 28-29: "…, and such information is critically needed in climate model for better estimating aerosol climate effects (Adebiyi and Kok, 2020)."

Lines 54-56: "However,. . .are lacking." Is this the first study on elements sampled in Antarctica? If yes, state clearly, if not, briefly mention about the prior studies that sampled aerosol in this region and what this study introduces. The paragraph that follows (lines 57-65) also does not give much idea about the knowledge gap that this study fills in.

> ***Response:*** We agree. We have changed the word "lacking" to "inadequate". Please see line 77. We also added a short summary on what have been done in Antarctic Peninsula and what has not been measured in lines 53-58.

> ***Modified text:***
> line 77: "…and accurate estimation of the atmospheric deposition of trace elements to the region are inadequate."
>
> lines 53-58: "In the Antarctic Peninsula, the concentrations of aerosol trace elements were measured at several sites (Dick, 1991; Artaxo et al., 1992; Mishra et al., 2004; Préndez et al., 2009). Total dust deposition in this region was also estimated based on the ice-core record (McConnell et al., 2007). However, the measurement of particle size distribution of aerosol trace elements in Antarctic Peninsula is missing and there is no direct measurement that evaluated the importance of atmospheric deposition as a source of nutrients for primary producers in West Antarctic Peninsula shelf waters."

Section 2, Line 66 onward: No mention of how the samples below "limits of detection (LOD)" were treated.

> ***Response:*** The concentrations below LOD were given a concentration of zero for the purposes of this study. We have added this description in the manuscript. Please see lines 129-130.

> ***Modified text:*** lines 129-130: "The concentrations below LOD were given a concentration of zero for the purposes of this study."

Lines 77-78: ". . .wind direction inside the sector ±60° from the direction of the station and wind speed <2 m s-1." Unclear to me.

> ***Response:*** Please see the plot below. When the wind came from the ±60° from the direction of Palmer Station or wind speed < 2 m s$^{-1}$, we paused the sampling to avoid local contamination from activities at the station. We have changed "and" to "or" in this sentence. Please see line 101.

[Figure]

> ***Modified text:*** line 101: "…inside the sector ±60° from the direction of the station buildings or when wind speed <2 m s$^{-1}$.

Line 91: ". . .Al, P, Ca, Ti, V, Mn, Ni, Cu, and Zn" this list looks different from the one mentioned in the abstract (line 9) that states ". . .Al, P, Ca, Ti, V, Mn, Ni, Cu, Zn, Ce, and Pb...".

> ***Response:*** This error has been corrected. Please see line 116.

> ***Modified text:*** line 116: "Elemental concentrations were determined for Al, P, Ca, Ti, V, Mn, Ni, Cu, Zn, Ce, and Pb."

Line 133: The summation is over what variable?

> ***Response:*** The summation refers to the sum of the concentrations determined for the 10 stages. We have revised the equation to mark i ranges from 1 to 10. Please see the equation in line 170.

Line 135: So, Vd was calculated from the combination of two models?

> ***Response:*** Yes.

Line 161: "The values of EFcrust for Ti, V, Mn, and Ce in aerosol samples were less than 10. . ." Why P is missing from this list where it has EFcrust<10 (Figure 3)?

> ***Response:*** The $EF_{crust}$ of P ranged from 2 to 8, slightly different from other crustal

elements. In addition, P possibly has a biogenic source that is also different from Ti, V, Mn, and Ce. Thus, we decided to discuss the P concentration and its source separately in the second paragraph in section 3.1.1.

Lines 170-171: "A similar phenomenon was observed at McMurdo Station where lightweight fuel oil was used that was not a significant source of V" not clear what was referred here.

> ***Response:*** We refer to the fact that light oil used at McMurdo Station was not a significant source of aerosol V at that location (Lowenthal et al., 2000); such situation could be true for aerosol V at Palmer Station, and our $EF_{crust}$ of V suggests that crustal source is the dominate source for V. Please see lines 210-211.

> ***Modified text:*** lines 210-211: "Similarly, unenriched V was observed at McMurdo Station where light-weight fuel oil was used that was not a significant source of V (Lowenthal et al., 2000)."

Lines 173-185: My understanding is that the purpose of this paragraph is to establish that the enrichment of P is higher than other crustal elements. But based on the results (Figure 3), it appears that P enrichment is not significantly high in this study, compared to the other relevant studies.

> ***Response:*** Yes, P was not enriched in our samples. We think P is special in this region since Antarctic Peninsula is one of the places that have the highest P excretion by seabird colonies (Otero et al., 2018). It's worthy to discuss the potential source of aerosol P and explain the reason why $EF_{crust}$ was relatively high in a separate paragraph.

Lines 197-198: "Hence, despite the recent increase in tourist ship traffic, it looks that Palmer Station was barely impacted by ship emissions" assuming this is correct, what is the reason for large variations of Ni (e.g., Figure 3)?

> ***Response:*** The significant variation of Ni from sample to sample might be attributed to receiving the long-range transport from South America. Please see line 287-289. In addition, the Ni has relatively high percentage of blank as shown in Table S1, and the variation in samples could also be contributed by the variation in the blank.

> ***Modified text:*** lines 287-289: "From the air mass back trajectories, the samples with high Ni (M2, M10), Cu (M1, M2, M4, M10) and Zn (M5) all were impacted by significant amount of air masses from the South Pacific Ocean and South America (Figure 5)."

Lines 199-200: "Ca accounts for about 3.5% of the weight of Earth's crust, while Ca is also a conservative major ion in seawater" provide citation.

> ***Response:*** Done. We have added Taylor and McLennan (1995) as the reference for the

Earth's crustal abundance and Millero (2016) as the reference for the major ions in seawater. Please see lines 237-238.

   ***Modified text:*** lines 237-238: "Ca accounts for about 3.5% of the weight of Earth's crust (Taylor and McLennan, 1995), while Ca is also a conservative major ion in seawater (Millero, 2016)…"

Line 215: ". . .the impact of the nearby McMurdo Dry Valleys" I could not understand what impact was referred here.

   ***Response:*** McMurdo Dry Valleys (MDVs) are dry lands in Antarctica. Consequently, the average Al concentration in $PM_{10}$ samples collected at McMurdo Station impacted by the air from MDVs was more than an order of magnitude higher than Antarctic Peninsula region (Mazzera et al., 2001). We have added a sentence to show this information for clarity. Please see lines 252-253.

   ***Modified text:*** lines 252-253: "The nearby McMurdo Sound was reported as the dustiest site in Antarctica (Winton et al., 2016)."

Lines 217-218: "The concentrations of Ti and Mn ranged from. . .respectively." I am a little confused if the ranges are for individual elements or both elements combined.

   ***Response:*** We have rephrased this sentence. Please see line 255-256.

   ***Modified text:*** lines 255-256: "The concentrations of Ti ranged from 140 to 800 pg m$^{-3}$ with an average of 250 pg m$^{-3}$, while the concentration of Mn ranged from 17 to 44 pg m$^{-3}$ with an average of 30 pg m$^{-3}$."

Lines 220-223: ". . .but comparable to the concentrations. . . (Figure 4e and g)" is it applicable to both Ti and Mn? Figures 4e and 4g do not support this claim. Mn concentration at AP-OS is more than double of AP-PS (Figure 4g).

   ***Response:*** We have revised this part of the text. Please see line 259.

   ***Modified text:*** line 259: "However, these Ti and Mn values observed at Palmer Station were in the same magnitude with the…"

Line 232: ". . .the P values. . ." P concentrations?

   ***Response:*** It should be P concentrations. We have removed this sentence for clarification.

Line 235-236: "Comparing global aerosol P concentrations. . .as those over the Central Pacific Ocean (Chen, 2004)" provide the concentration values from the referred study.

*Response:* We realized this comparison is not necessary and have decided to remove it.

Lines 236-237: "Confirming that Palmer Station was little influenced by aerosols derived from biomass burning through long-range transport, the calculated non-sea-salt-K was indistinguishable from zero." I could not understand what this means.

*Response:* We revised this sentence. Please see lines 272-274.

*Modified text:* line 272-274: "In this study, nss-$K^+$ was used as a tracer of biomass burning (Winton et al., 2015). The calculated nss-$K^+$ was indistinguishable from zero, suggesting that $K^+$ in aerosol at Palmer Station was primarily derived from sea water, not from biomass burning through long-range transport."

Lines 239-241: ". . .suggesting that aerosol crustal elements observed at Palmer Station were impacted by sources in that region (Figure 5)." Why this argument applies to only crustal elements?

*Response:* In this section, we focused on discussing crustal elements. We want to address that the regional crustal sources play more important roles than the sources in distance.

Lines 263-264: "The low concentrations of heavy metals observed during this study suggest that local anthropogenic emissions were negligible." Which metals you are referring to?

*Response:* It refers to aerosol Pb. We have revised this sentence. Please see line 311-312.

*Modified text:* line 311-312: "The low concentrations of Pb observed in samples associated with air masses that did not pass over Southern South America suggest that local anthropogenic emissions were negligible."

Lines 264-265: "Thus the major source of non-crustal elements in aerosols over the study region may be long-range transport from regions impacted by anthropogenic emissions" very weak conclusion as it is only based on Pb variation.

*Response:* We agree. We have added a short discussion to show that the high concentrations of Ni, Cu, and Zn are also associated with air masses derived from coastal South America (lines 287-291). We also include a short summary for the potential sources of the anthropogenic elements in South America (lines 296-299).

*Modified text:*
lines 287-291: "From the air mass back trajectories, the samples with high Ni (M2, M10), Cu (M1, M2, M4, M10) and Zn (M5) all were impacted by significant amount

of air masses from the South Pacific Ocean and South America (Figure 5). The back trajectories of air masses of M7 didn't touch South America but the concentration of Zn in this sample was high. With the fact that aerosol Zn was found in both fine- and coarse-mode fractions (Table 2), both local sources and long-range transport may contribute to this element in the air."

lines 296-299: "In South America, high enrichment of Ni, Cu, Zn, and Pb in fine mode particles was reported to be primarily associated with vehicle emission, soil dust, and oil combustion (Artaxo et al., 1999; Jasan et al., 2009). Moreover, miming activities were suggested as an important source, especially in remote sites in South America (Carrasco and Préndez, 1991; Klumpp et al., 2000)."

Line 271: ". . .27 and 26, in seawater" respectively?

*Response:* Yes. We have repaired this sentence. Please see lines 321-323.

*Modified text:* lines 321-323: "Thus, the $Na^+/K^+$ ($32 \pm 3.5$) ratios was close to the average Na/K mass ratio in seawater (27) and the $Na^+/Ca$ ratios ($31 \pm 5.5$) were close to the average $Na^+/Ca$ ratio in seawater (26) as well (Millero, 2016)."

Lines 272-273: "The results suggest that Ca was dominated by sea-salt aerosol. . ." what about K?

*Response:* We have added that $K^+$ was primarily derived from sea-salt aerosol as well, in the Discussion. Please see lines 324-325.

*Modified text:* lines 324-325: "Therefore, the Ca and K+ in aerosols were derived primarily from sea salt at Palmer Station."

Line 276: "classified into three groups based on their potential dominant sources" is this classification is in the context of section 3.1? If yes, it should be stated clearly. If that's not the case, provide justification of the grouping.

*Response:* We have revised the manuscript and added k-means clustering to classify the size distributions into 5 clusters: (1) crustal elements from crustal weathering and wind-induced resuspension of soil particles, (2) Al dominated by local minerals, (3) Pb from anthropogenic sources, as a result of long-range transport, (4) sea salt elements from the ocean, through bursting bubbles of seawater, and (5) P from local biogenic and soil resuspension. Please see lines 330-332. We also add the clustering method in "Method". Please see section 2.3.2, lines 161-163.

*Modified text:*
lines 330-332: "…with each group showing a unique size distribution pattern: (1) crustal elements from crustal weathering and wind-induced resuspension of soil

particles, (2) Al dominated by local minerals, (3) Pb from anthropogenic sources, as a result of long-range transport, (4) sea salt elements from the ocean, through bursting bubbles of seawater, and (5) P from local biogenic and soil resuspension."

lines 161-163: "The k-means clustering algorithm was used to cluster the average particle size distribution of each trace element. The optimal number of clusters (k) was selected by choosing the k with the highest Calinski-Harabasz index (Caliński and Harabasz, 1974)."

Lines 301-302: "The mass distributions of sea-salt elements (Ca, Na and K) as the third group were dominated by coarse-mode particles with diameters 2.5–7.8 μm (Figure 6)" I could not find the size distributions of Na and K in Figure 6!

*Response:* We added a new figure, the original Figure 6 now become Figure 7 and Figure 8. We have added the particle size distribution of $Na^+$ and $K^+$ in Figure 7.

Line 303: ". . .the correlation between the total concentrations of Ca and Na was strongly positive ($R2\pm0.82$, p-value < 0.01). . ." was it based on the 8 pairs of samples presented in Table 3? What about the correlations of other elements (such as K) with Na? Like Ca, If K is also associated with seasalt (as suggested in lines 270-272) one would expect a good K-Na correlation.

*Response:* Yes, the correlation between total concentrations of Ca and Na was based on 8 pairs of samples in Table 3. As the reviewer predicted, the correlation between Na and K was even better with a $R^2 = 0.96$. We have revised this section and removed this part.

Lines 311-313: "The rough estimates of the dry deposition fluxes of Ni, Cu, and Zn at Palmer Station. . ." should mention few values from the literature so the readers get an idea of how large or small the values are.

*Response:* We have showed the dry deposition fluxes of Cu and Zn measured in the North Atlantic Ocean for comparison. Please see lines 403-406.

*Modified text:* lines 403-406: "The rough estimates of the dry deposition fluxes of Ni and Zn at Palmer Station are close to the median deposition fluxes found in the western North Atlantic Ocean (Ni: 18 μg m$^{-2}$ yr$^{-1}$, Zn: 16 μg m$^{-2}$ yr$^{-1}$), whereas the dry deposition flux of Cu is slightly higher than the median Zn dry deposition flux (2.8 μg m$^{-2}$ yr$^{-1}$) in the western North Atlantic Ocean (Shelley et al., 2017)."

Lines 313-315: "The estimated dry deposition fluxes of total continental dust. . . among the lowest globally (Lawrence and Neff, 2009)" should mention a few global values.

*Response:* Lawrence and Neff, 2009 provides an average dust deposition fluxes among

the areas receiving dust from local (0–10 km), regional (10–1000 km), and global (>1000 km) scales. For global scale at remote sites, the average dust deposition flux is 0.4 g m$^{-3}$ yr$^{-1}$.We have revised this sentence in lines 408-409.

> ***Modified text:*** lines 408-409: "…, and this fluxes is only around 10% of the mean global dust deposition flux at remote sites (Lawrence and Neff, 2009)."

Lines 316-319: ". . .precipitation scavenging accounted for about 40-60% of the total deposition. . ." are these fractions yearly average? Contributions of wet deposition to the total flux is a strong function of season. This study is limited to only two months of sampling. Authors need to discuss the likely uncertainties involved with extrapolating the short-term dry or wet deposition flux to yearly contributions.

> ***Response:*** The range of 40-60% is a result from a modelling study. We are aware that the precipitation conditions and the proportion of wet deposition in total deposition could differ significantly in different seasons. However, due to the lack of previous measurement of dry and wet deposition fluxes in this region, it's hard to evaluate the uncertainties.

Lines 318-319: "Assuming this wet deposition fraction applies to the Antarctic Peninsula region, we approximate roughly a total dust flux of 10 mg m-2 yr-1" I understand that this is an estimation, but how was it obtained? Assuming dry/total as 0.4? or 0.6?

> ***Response:*** Yes. We applied the ratio of dry deposition/total deposition (0.4 and 0.6) to the dry deposition fluxes of 5.5 mg m$^{-2}$ yr$^{-1}$. The result ranged from 9.2 to 14 mg m$^{-2}$ yr$^{-1}$, which is around 10 mg m$^{-2}$ yr$^{-1}$. We have added the ratio (0.4 and 0.6) in this sentence to make it clear. Please see line 413.

> ***Modified text:*** line 411: "Assuming this wet deposition fraction (0.4-0.6) applies…"

Lines 329-341: Authors need to carefully revise this entire paragraph to ensure it is readable.

> ***Response:*** We have revised and added more information in this paragraph, and please see lines 427-441.

> ***Modified text:*** Please see the paragraph discussing the importance of atmospheric input to the particulate element concentrations in surface seawater of the West Antarctic Peninsula in lines 427-441.

Lines 346-348: "Most of the samples collected during this study were impacted by air masses originating around or passing over Northern Antarctic Peninsula. . ." This claim is not supported by any strong evidence. Airmass trajectories associated with 4 of the total samples were presented in Figure 5.

***Response:*** We have rewritten the Conclusions and Implications section. This statement has been changed to a conclusion that local/regional sources contributed to the concentrations of crustal elements. Please see lines 446-448. We have also added air mass trajectories for all samples.

***Modified text:*** lines 446-448: "The particle size distributions of crustal elements, including Al, P, Ti, V, Mn, and Ce, were all concentrated in coarse mode, suggesting strong regional emissions likely from ice-free areas on the Antarctic Peninsula and its associated islands."

Lines 354-357: "As the role of wet deposition is unquantified at. . .may need to be reevaluated." I could not understand what this sentence is referring to.

***Response:*** In this study, we only obtained the estimation of dry deposition fluxes, while wet deposition remains unknown. If the future measurements show that wet deposition flux in this region accounts for far more or less than the assumption we made, the total deposition flux of dust need to be reevaluated. We have revised this sentence to make it clear. Please see lines 456-460.

***Modified text:*** lines 456-460: "As the role of wet deposition is unquantified at present and remains poorly constrained for this region, the total deposition fluxes of trace elements during the austral summer could exceed the dry deposition fluxes reported here. Therefore, the importance of atmospheric deposition of trace elements to coastal West Antarctic Peninsula may need to be re-evaluated with additional observations of wet deposition.

Figure 2: Missing proper x-axis label. In addition, the figure caption should mention what the legends (M1. . .M10) are referring to.

***Response:*** We have revised the x-axis label and figure caption. Please see revised Figure 2.

Figure 4: This figure should be revised.
1) What is AP-OS1 shown in Figure 4(g)?
2) What are the vertical lines representing?
3) The acronyms of the sites mentioned in the caption should be consistent with the ones shown on figures (e.g., AP-PS or PS).
4) As mentioned in the text (e.g., line 211), air mass samples from many of the previous studies used for the comparison correspond to PM10 or even PM2.5. The figures should clearly indicate this (e.g., AP-OS(PM2.5) ).

***Response:*** (1) We have corrected this typo. It should be "AP-OS". (2) The vertical lines represent the standard deviation of the trace element concentrations in each study. We have included this information in the revised figure caption. (3) We have checked and

revised the incorrect acronyms. We also mark our study in red in the revised Figure 4. (4) We have marked PM$_{2.5}$ and PM$_{10}$ samples in the figure caption. Please see the revised Figure 4.

Figure 5: Why is this figure showing 4 samples only?

***Response:*** We have included the back trajectories for all 8 samples. Please see the revised Figure 5.

Table 4: Should include a few extra columns showing literature values.

***Response:*** Our estimated dry deposition fluxes of trace elements are much lower than the other sites, and we have briefly compared our estimates with literature values in the text. Please see lines 403-406.

***Modified text:*** lines 403-406: "The rough estimates of the dry deposition fluxes of Ni and Zn at Palmer Station are close to the median deposition fluxes found in the western North Atlantic Ocean (Ni: 18 µg m$^{-2}$ yr$^{-1}$, Zn: 16 µg m$^{-2}$ yr$^{-1}$), whereas the dry deposition flux of Cu is slightly higher than the median Zn dry deposition flux (2.8 µg m$^{-2}$ yr$^{-1}$) in the western North Atlantic Ocean (Shelley et al., 2017)."

**References**

Adebiyi, A. A., and Kok, J. F.: Climate models miss most of the coarse dust in the atmosphere, Science Advances, 6, eaaz9507, 2020.

Artaxo, P., Rabello, M. L., Maenhaut, W., and GRIEKEN, R. V.: Trace elements and individual particle analysis of atmospheric aerosols from the Antarctic Peninsula, Tellus B, 44, 318-334, 1992.

Artaxo, P., Oyola, P., and Martinez, R.: Aerosol composition and source apportionment in Santiago de Chile, Nuclear Instruments and Methods in Physics Research Section B: Beam Interactions with Materials and Atoms, 150, 409-416, 1999.

Caliński, T., and Harabasz, J.: A dendrite method for cluster analysis, Communications in Statistics-theory and Methods, 3, 1-27, 1974.

Carrasco, M. A., and Préndez, M.: Element distribution of some soils of continental Chile and the Antarctic peninsula. Projection to atmospheric pollution, Water, Air, and Soil Pollution, 57, 713-722, 1991.

Dick, A.: Concentrations and sources of metals in the Antarctic Peninsula aerosol, Geochimica et cosmochimica acta, 55, 1827-1836, 1991.

Jasan, R., Plá, R., Invernizzi, R., and Dos Santos, M.: Characterization of atmospheric aerosol in Buenos Aires, Argentina, Journal of Radioanalytical and Nuclear Chemistry, 281, 101-105, 2009.

Klumpp, A., Domingos, M., and Pignata, M. L.: Air pollution and vegetation damage in South America—state of knowledge and perspectives, CRC Press LLC, United States of America, 2000.

Lawrence, C. R., and Neff, J. C.: The contemporary physical and chemical flux of aeolian dust:

A synthesis of direct measurements of dust deposition, Chemical Geology, 267, 46-63, 2009.

Lowenthal, D. H., Chow, J. C., Mazzera, D. M., Watson, J. G., and Mosher, B. W.: Aerosol vanadium at McMurdo Station, Antarctica:: implications for Dye 3, Greenland, Atmospheric environment, 34, 677-679, 2000.

Mazzera, D. M., Lowenthal, D. H., Chow, J. C., Watson, J. G., and Grubĭsíc, V.: PM10 measurements at McMurdo station, Antarctica, Atmospheric Environment, 35, 1891-1902, 2001.

McConnell, J. R., Aristarain, A. J., Banta, J. R., Edwards, P. R., and Simões, J. C.: 20th-Century doubling in dust archived in an Antarctic Peninsula ice core parallels climate change and desertification in South America, Proceedings of the National Academy of Sciences, 104, 5743-5748, 2007.

Millero, F. J.: Chemical oceanography, CRC press, 2016.

Mishra, V. K., Kim, K.-H., Hong, S., and Lee, K.: Aerosol composition and its sources at the King Sejong Station, Antarctic peninsula, Atmospheric Environment, 38, 4069-4084, 2004.

Nelson, P.: The James Ross Island Volcanic Group of north-east Graham Land, British Antarctic Survey, 1966.

Pereira, P. S., van de Flierdt, T., Hemming, S. R., Hammond, S. J., Kuhn, G., Brachfeld, S., Doherty, C., and Hillenbrand, C.-D.: Geochemical fingerprints of glacially eroded bedrock from West Antarctica: Detrital thermochronology, radiogenic isotope systematics and trace element geochemistry in Late Holocene glacial-marine sediments, Earth-Science Reviews, 182, 204-232, 2018.

Préndez, M., Wachter, J., Vega, C., Flocchini, R. G., Wakayabashi, P., and Morales, J. R.: PM2.5 aerosols collected in the Antarctic Peninsula with a solar powered sampler during austral summer periods, Atmospheric Environment, 43, 5575-5578, 10.1016/j.atmosenv.2009.07.030, 2009.

Shelley, R. U., Roca-Martí, M., Castrillejo, M., Sanial, V., Masqué, P., Landing, W. M., van Beek, P., Planquette, H., and Sarthou, G.: Quantification of trace element atmospheric deposition fluxes to the Atlantic Ocean (> 40 N; GEOVIDE, GEOTRACES GA01) during spring 2014, Deep Sea Research Part I: Oceanographic Research Papers, 119, 34-49, 2017.

Taylor, S. R., and McLennan, S. M.: The geochemical evolution of the continental crust, Reviews of geophysics, 33, 241-265, 1995.

Winton, V. H. L., Bowie, A. R., Edwards, R., Keywood, M., Townsend, A. T., van der Merwe, P., and Bollhöfer, A.: Fractional iron solubility of atmospheric iron inputs to the Southern Ocean, Marine Chemistry, 177, 20-32, 10.1016/j.marchem.2015.06.006, 2015.

Zoller, W. H., Gladney, E., and Duce, R. A.: Atmospheric concentrations and sources of trace metals at the South Pole, Science, 183, 198-200, 1974.

---

## Author Comment (AC1)

**Major Comments:**

Fan et al. report size segregated aerosol concentrations and dry deposition fluxes for a suite of trace elements and cations from a land-based station on the Antarctic Peninsula over the 2016-2017 austral summer. Aerosol iron solubility data from these samples has been previously published by Gao et al. (2020). While the timeseries is short (8 sets of size segregated samples each with a sampling duration of 1 week), the study fills a gap in data availability of aerosol trace element levels in this sector of Antarctica and also in our understanding of the deposition pathways. The authors have estimated dry deposition velocities for each trace element at the sampling site, providing a more reliable estimate of dry deposition fluxes than previous studies which typically use a generic value not necessarily appropriate for the region with a large uncertainty. The study also reports aerosol size distribution of the trace elements which is extremely valuable data, also scarce in this region, providing important insights into the source and transport pathways of aerosols in pristine airmasses. The manuscript is well written, has a logical structure and provides important data to a wide-ranging community of researchers from atmospheric chemists, biogeochemists and ice core scientists. I recommend publication in ACP after a few suggestions for improvement have been addressed. I hope the suggestions below will strengthen the manuscript.

> **_Response:_** We appreciate Dr. Winton's evaluation of this study and agree that the revision is needed along the lines of her suggestions.

**Suggestions for improvement:**

*Introduction:* It would be helpful to add a short section on what is known about aerosol removal processes in the Southern Ocean and where the gaps are in our understanding. I also think it would be worth briefly summarizing the aerosol iron data here e.g. mineral dust source and probable source region.

> **_Response:_** We agree with Dr. Winton. A brief explanation about aerosol removal mechanisms and current gaps in our understanding about aerosols over the Southern Ocean and Antarctica has been added in lines 29-39. A short summary of ice-free areas that may be potential dust source regions on the Antarctic Peninsula have been added in lines 62-66. A brief summary about what we found in our previously published Fe-focused study is added in lines 66-69.

*Modified text:*

lines 29-39: "In the atmosphere, the removal of aerosols involves gravitational settling, impaction, diffusion, hygroscopic growth, and scavenging by precipitation, and the rates of all these processes are dependent on the aerosol particle size (Saltzman, 2009). Over the Southern Ocean and Antarctica, aerosol particle size distributions have been studied (Gras, 1995; Järvinen et al., 2013; Xu et al., 2013; Kim et al., 2017; Herenz et al., 2019; Lachlan-Cope et al., 2020). Seasonal variations of the particle number concentrations were observed in the King Sejong Station, Antarctic Peninsula, and Halley Research Station on the Brunt Ice Shelf. The maximum and minimum of the particle number concentrations at two sites were found in the austral summer and austral winter, respectively (Kim et al., 2017; Lachlan-Cope et al., 2020). Due to the low background concentrations of aerosol particles, new particle formation has been suggested to substantially affect the annual aerosol concentration cycles (Lachlan-Cope et al., 2020). However, most of these studies focused on the physical characterises of aerosol particle size; the size distributions of aerosol trace elements are still poorly understood, and at present only aerosol Fe has been characterized for particle-size distributions around Antarctica (Gao et al., 2013, 2020)."

lines 62-66: "Under conditions of low precipitation (Van Lipzig et al., 2004) and high wind speed (Orr et al., 2008), several ice-free areas on James Ross Island, off the east coast of the northern Peninsula could serve as local dust sources (Kavan et al., 2018), and such sources may contribute to the atmospheric loading of certain trace elements such as Fe (Winton et al., 2014; Gao et al., 2020). Similar ice-free areas were found in King George Island, Livingston Island, Anvers Island etc. in the Antarctic Peninsula region (Bockheim et al., 2013), and may act as potential sources of aeolian dust."

lines 66-69: "As part of the current study, the particle size distribution of aerosol Fe measured at Palmer Station showed single mode distribution and was dominated by coarse particles, suggesting that local regional dust emission dominated the concentration of Fe in this region (Gao et al., 2020)."

*Discussion:* The discussion on aerosol sources and, in particular, particle size distribution needs clarification and further investigation. The particle size distribution is a great dataset but at the moment it hasn't been fully utilized. Incomplete referencing and a limited explanation of the particle distribution let the discussion down. The authors could improve this section with a statistical cluster analysis of the physical characteristics of particle size distributions and/or a thorough understanding/comparison of the Antarctic and Southern Ocean literature on this topic. Currently, there is no discussion on what the size distribution tells us about the atmospheric transport mechanisms of the individual trace elements. Also, the conclusions don't come through strongly enough in the discussion.

*Response:* We agree with Dr. Winton. The discussion on the particle size distribution has been rewritten. We used k-means clustering to classify the normalized particle size distribution of trace element/ionic species into 5 clusters to further discuss the primary

source of each element: (1) mineral dust (Ti, V, Mn, Ce), (2) Al, (3) Pb, (4) sea salt (Ca, Na$^+$, K$^+$), and (5) P. Trace elements were grouped by their particle size distributions and then discussed with respect to potential sources. Samples that showed unique size distribution pattern were discussed individually. In this part, we also added a comparison of aerosol size distributions with studies focused on the dust particle size distributions in snow, ice core, and aerosols in Antarctica. Please see the section 3.3, lines 326-395. The method of k-means clustering has been added to section 2.3.2, lines 155-163.

***Modified text:***
lines 326-395: Please see section 3.3.

lines 155-163: Please see section 2.3.2.

***Conclusions and implications:*** It's fantastic to see the size distribution data. Do you have plans to continue aerosol monitoring at this land-based sampling station to make a long-term record of size segregated trace elements? In this section, you could recommendation future aerosol studies investigate the size distribution to provide additional information on the source and atmospheric transport. What are the implications for the underestimation and overestimation of dry deposition velocities in previous aerosol flux estimates and climate modelling for the region? What about the implications of your new particle size distribution data for new particle formation studies?

> ***Response:*** We have revised the conclusions and implications. Please see the revised section 4, lines 442-463. We are hoping to continue the measurements of aerosol chemistry in this region through future studies, particularly through long-term observations. The errors associated with the dry deposition velocities could cause a biased dry deposition flux estimation and mislead modeling results. The range of particle sizes (from ~0.1 – 10 µm) in our study is much larger than on the sizes critical for new particle formation (in the subnanometer – nanometer range), and thus it is difficult to address the new particle formation processes with our data. We cannot say much about the new particle formation process.

> ***Modified text:*** lines 442-463: Please see section 4.

***Throughout:*** Please make it clear throughout the study that you are reporting total acid digestible trace element concentrations and water soluble cations. Please report Ca and K as ions Ca2+ and K+ rather than elements throughout the manuscript.

> ***Response:*** All elements except for Na+ and K+ were reported for the total acid digestible concentrations which were analyzed by ICP-MS. To avoid confusion, we have added charge to all water-soluble cations (Na$^+$ and K$^+$) in the text.

**Technical corrections:**

Line 11 Suggest replacing with "local mineral dust, long-range transported aerosol and sea salt aerosol." What about local P biogenic emissions?

> ***Response:*** We have revised this part by adding "regional crustal emissions, long-range transport and sea salt." We have briefly mentioned about local P biogenic emissions. Please see lines 12-13.

> ***Modified text:*** lines 12-13: "…these elements are derived primarily from three sources: (1) regional crustal emissions, including possible resuspension of soils containing biogenic P, (2) long-range transport, and (3) sea-salt.

Line 26 Define course mode.

> ***Response:*** We have revised this part of the text. Please see lines 28-29.

> ***Modified text:*** lines 28-29: "…, and such information is critically needed in climate model for better estimating aerosol climate effects (Adebiyi and Kok, 2020)."

Line 32 Include sea salt as a source of aerosol.

> ***Response:*** We have added sea salt as a source of trace elements (please see line 43).

> ***Modified text:*** line 43: "In addition, sea salt emission, …"

Line 37 A number of local dust sources in Antarctica are increasingly being quantified. See Delmonte et al. (2019) for a recent review. Delmonte, B., Winton, H., Baroni, M., Baccolo, G., Hansson, M., Andersson, P., Baroni, C., Salvatore, M.C., Lanci, L. and Maggi, V., 2020. Holocene dust in East Antarctica: Provenance and variability in time and space. The Holocene, 30(4), pp.546-558.

> ***Response:*** We have cited Kavan et al. (2018); Delmonte et al. (2020) and added the local dust emission as a source of atmospheric trace elements to the Southern Ocean and Antarctica in addition to long-range transport (please see lines 48-49).

> ***Modified text:*** lines 48-49: "Atmospheric trace elements over the Southern Ocean and Antarctica may derive from distant continental sources through long-range transport (Li et al., 2010) and local dust sources in certain areas (Kavan et al., 2018; Delmonte et al., 2020)."

Line 47 Include a refence to McMurdo Sound here as the dustiest region in Antarctica.

> ***Response:*** We have cited Winton et al. (2014) as a reference measuring dust in McMurdo Sound (please see line 64).

Line 62 ": : :new sources of aerosol trace elements by warming which exposes a greater area of ice-free land and by: : :".

> ***Response:*** We have revised this sentence (please see lines 83-85).
>
> ***Modified text:*** lines 83-85: "The new observational data also provide insight into sources of aerosol trace elements, as influenced recently by warming, which exposes a greater area of ice-free land, and by the impact of human activities in this region."

Line 68 Mention that a detailed description of the aerosol sampling and protocols to mitigate contamination in the pristine environment can be found in Gao et al. (2020).

> ***Response:*** We have added a sentence to indicate that the detailed sampling protocols can be found in Gao et al. (2020) (please see lines 91-93).
>
> ***Modified text:*** lines 91-93: "A detailed description of the aerosol sampling, including the protocols for mitigating contamination in the pristine environmental can be found in Gao et al. (2020)."

Line 100 Please report precision here or in Table 2.

> ***Response:*** We have moved Table 2 to Supplement as Table S1. The precisions were estimated by duplicate measurement and reported in Table S1. A sentence refers to Table S1 has been added in lines 124-125.
>
> ***Modified text:*** lines 124-125: "During the ICP-MS analysis, duplicate injections of sample solutions were made every ten samples to check the instrument precision (Table S1)."

Line 107 Did you measure other cations and anions on these samples? If you did, will these data be included in your US Antarctic Program Data Centre data publication? I understand they may not be relevant for this manuscript but I'm sure they would be beneficial to other researchers if the data is available. Why did you not measure Ca an K by ICP-MS?

> ***Response:*** We measured other cations and anions. Those data will be used for another paper and will be uploaded to the US Antarctic Program Data Center shortly. In this study, we measured Ca by ICP-MS; the water-soluble cations $Na^+$ and $K^+$ were measured by IC. The main reason for measuring those elements by IC is that in most cases, only water-soluble $Na^+$ and $K^+$ are used as tracers for sea-salt and biomass burning, respectively.

Line 121 Why did you select Al over Ti as the crustal reference?

***Response:*** Al has been used as the crustal reference by other investigators in many locations in Antarctica, such as the Antarctic Peninsula (Dick, 1991), McMurdo (Lowenthal et al., 2000), the South Pole (Zoller et al., 1974). Therefore, our results can be compared directly with others on the same basis. We have added these citations in the revised manuscript (please see lines 151-153).

***Modified text:*** line 151-153: "The crustal reference element Al has been widely used to calculate crustal enrichment factors in the Southern Ocean and Antarctica (Zoller et al., 1974; Dick, 1991; Lowenthal et al., 2000; Xu and Gao, 2014)."

Line 128 ": : :was calculated from the concentration of the trace element in the air and dry: : :"

***Response:*** We have revised this sentence (please see lines 165-166).

***Modified text:*** lines 165-166: "Dry deposition flux ($F_d$, mg m$^{-2}$ yr$^{-1}$) of each element in aerosols was calculated from the concentration ($C_e$, ng m$^{-3}$) of the trace element in the air and dry deposition velocity ($V_d$, cm s$^{-1}$):"

Line 141 What are the environmental factors?

***Response:*** For coarse particles, gravitational settling dominates the dry deposition velocity. In contrast, wind speed, relative humidity, air temperature, and sea surface temperature were considered as the environmental factors that control the dry deposition velocity of fine particles, although these factors also affect the dry deposition of coarse-particles. We have added those factors in the revised manuscript (please see lines 178-179).

***Modified text:*** lines 178-179:"…, including wind speed, relative humidity, air temperature, and sea surface temperature (Figure 2)"

Line 142 Please report the uncertainty of your dry deposition velocity estimates. Also, where can the reader find the dry deposition velocity values for each element?

***Response:*** We have reported the uncertainties in the revised manuscript; please see lines 179-181. The values of dry deposition velocity for each element is reported in Table 3.

***Modified text:*** lines 179-181: "Therefore, the uncertainties of the dry deposition velocity estimation for the elements that were dominated by coarse particles were about ±30-60%, and the uncertainty of fine particle dominated element (Pb) was about ±100%."

L160-185 Can you use refences for P and V sources from the Southern Hemisphere? I'm not aware of any evidence that agricultural and industrial emissions from China are transported to Antarctica. It seems that P at your site is dominantly sourced from local Antarctic emissions

rather than long-range transported agricultural or volcanic emissions. Do airmasses pass over Antarctic soils, or seabird colonies? While ship emissions of V have been detected in aerosols in the Arctic and Atlantic where shipping is more frequent, is there any evidence of this in the Southern Ocean? While it's important to identify all possible sources of these elements, this section could be tightened up to clarify the most likely sources of this group of elements and rule out the unlikely ones to Antarctica.

**_Response:_** We have searched the related references for V in the Southern Hemisphere and found only one study conducted in New South Wales, Australia, using V/Ni as a tracer of the use of heavy oils (Keywood et al., 2020). We have replaced the previous references with this one. Please see line 209. In addition, we didn't find any previous studies that reported an elevated $EF_{crust}$ of V due to the ship emissions in the Southern Ocean or coastal Antarctica. It looks like that ship emissions may not be an important source for V in this region, similar to the results found at McMurdo (Lowenthal et al., 2000).

For P, we have removed biomass burning as a source of P since the nss-$K^+$ suggests our samples were barely affected by biomass burning. The 72-hour air mass trajectories show most of the air masses passing over the Antarctic Peninsula were derived from the South Pacific Ocean or South Atlantic Ocean. However, there is a penguin colony on Torgerson Island (~ 1 km from Palmer Station), and it is possible that our samples were affected. In addition, the Antarctic Peninsula has one of the highest P excretions by seabird colonies globally (Otero et al., 2018) (Please see the figure below). We believe that the local crustal emissions from the soil can definitely emit P-enriched dust particles to the atmosphere.

[Figure]

Global distribution of N and P excretion by seabird colonies (Otero et al., 2018).

**_Modified text:_** line 209: "…and V in aerosols was found associated with ship emissions due to the use of heavy oil fuel (Keywood et al., 2020)."

Line 186 and 242 Many of these elements occur naturally in the crust but in your samples they

have a non-crustal source. I wonder if you could rename the headings to avoid confusion e.g. rename the group of elements into one of the three categories described in the abstract - local mineral dust, long-range transported aerosol and sea salt aerosol.

> ***Response:*** We have changed the subtitles in "3.2 Concentrations of trace elements" to "mineral dust", "anthropogenic aerosol", and "sea salt aerosol". Please see section 3.2. However, in the "3.1 Enrichment factors of trace elements" section, the $EF_{crust}$ can only tell us if the elements were derived from crustal or non-crustal sources. We prefer to keep crustal source and non-crustal source.

Line 195 Again, please use refences for the Southern Ocean rather than Northern Hemisphere. Some suggestions for Pb: McConnell, J.R., Maselli, O.J., Sigl, M., Vallelonga, P., Neumann, T., Anschütz, H., Bales, R.C., Curran, M.A., Das, S.B., Edwards, R. and Kipfstuhl, S., 2014. Antarctic-wide array of high-resolution ice core records reveals pervasive lead pollution began in 1889 and persists today. Scientific Reports, 4(1), pp.1-5. Bollhöfer, A.F., Rosman, K.J.R., Dick, A.L., Chisholm, W., Burton, G.R., Loss, R.D. and Zahorowski, W., 2005. Concentration, isotopic composition, and sources of lead in Southern Ocean air during 1999/2000, measured at the Cape Grim Baseline Air Pollution Station, Tasmania. Geochimica et Cosmochimica Acta, 69(20), pp.4747-4757.

> ***Response:*** We have replaced all the references with Dick (1991), Artaxo et al. (1992), McConnell et al. (2014), and Keywood et al. (2020), which were all conducted in the Antarctica or Southern Hemisphere. Please see lines 228-229 and line 233.

Line 200-201 This sentence isn't clear whether you are suggesting that Ca is derived from seawater or another source? For the modern day, many coastal ice core sites suggest a marine source of Ca, where during glacial periods Ca is dominated by a long-range transported mineral dust.

> ***Response:*** We have rephrased this sentence. Palmer Station should receive Ca from both seawater and crustal emission. In this study, the high crustal enrichment factor (>10) of Ca suggests the concentration of Ca was dominated by sea-salt aerosol. Please see line 238.

> ***Modified text:*** line 238: "The high $EF_{crust}$ for Ca at Palmer Station suggests that aerosol Ca was mainly derived from sea-salt."

Line 208-211 Refer the reader to Fig. 4.

> ***Response:*** We have referred this sentence to Figure 4. Please see line 250.

> ***Modified text:*** line 250: "…the 5-year summer mean of 1.3 ng m-3 in East Antarctica (Weller et al., 2008) and the summer average of 0.57 ng m-3 measured at the South Pole (Zoller et al., 1974; Maenhaut et al., 1979) (Figure 4b)"

Line 215-216 Al concentrations have been measured in local dust in snow on sea ice in McMurdo Sound which is the dustiest location in Antarctica.

> ***Response:*** We have added a sentence that refers McMurdo Sound is the dustiest location in Antarctica and cite Winton et al. (2016) in the revised manuscript. Please see lines 252-253.

> ***Modified text:*** lines 252-253: "The nearby McMurdo Sound was reported as the dustiest site in Antarctica (Winton et al., 2016)."

Line 232-235 Comparison to trace element concentrations in regions outside of the Southern Ocean or the Southern Hemisphere seems irrelevant.

> ***Response:*** We have removed the comparison with other studies conducted in the Northern Hemisphere.

L235-237 These pieces of evidence further confirm your EF interpretation that P was mainly locally derived from soil and bird colonies proximal to the station. How did you calculate nss-K? Please state the limitations of using nss-K as a biomass burning tracer in the Southern Ocean as opposed to sites proximal to biomass/fires. See other studies in the Southern Ocean e.g. nss-K at Cape Grim. Is there a better tracer of biomass burning that you have access to over the sampling period? Winton, V.H.L., Bowie, A.R., Edwards, R., Keywood, M., Townsend, A.T., van der Merwe, P. and Bollhöfer, A., 2015. Fractional iron solubility of atmospheric iron inputs to the Southern Ocean. Marine Chemistry, 177, pp.20-32.

> ***Response:*** The nss-$K^+$ was calculated using the equation: [nss-$K^+$]=[$K^+$]-0.037[$Na^+$], where 0.037 is the mass ratio between K and Na in seawater (Millero, 2016). We have added this equation to the methods. Please see line 138.
> The main disadvantage of using nss-$K^+$ as the tracer of biomass burning is that water soluble K can have multiple sources, which include not only biomass burning and sea-salt, but also mineral dust. During long-range transport, those sources may contribute K to the air mass and cause an increase in nss-$K^+$. In our samples, the nss-$K^+$ were close to 0. Such a result means $K^+$ were dominated by sea-salt aerosol and all the other sources, including biomass burning, played a minor role. We are aware there are some better tracers, such as levoglucosan. However, we do not have enough samples for further analysis. We have revised the sentence and added Winton et al. (2015) as a reference that use nss-$K^+$ as a tracer of biomass burning in lines 272-274.

> ***Modified text:*** lines 272-274: "In this study, nss-K+ was used as a tracer of biomass burning (Winton et al., 2015). The calculated nss-K+ was indistinguishable from zero, suggesting that K+ in aerosol at Palmer Station was primarily derived from sea water, not from biomass burning through long-range transport."

Line 243-255 Open paragraph with a sentence to let the reader know you are discussing Ni, Zn, Pb and Cu first. It's not clear what caused the large variability in this group of elements (Ni, Zn, Pb and Cu). Can you link these elements with air mass trajectories as you did with Pb? What anthropogenic activities emit Ni, Zn, Pb and Cu in South America?

***Response:*** We have added a sentence in the beginning of this paragraph (please see line 280). A short discussion about the link between air mass back trajectories and high concentrations of Ni, Cu, and Zn were as added to this part (please see line 287-291). In South America, the emissions of Ni, Zn, Pb, and Cu were suggested to be primarily associated with vehicle emission, soil dust, and oil combustion (Artaxo et al., 1999; Jasan et al., 2009). In addition, mining activities were suggested as an important anthropogenic source at remote sites in the South America (Carrasco and Préndez, 1991; Klumpp et al., 2000). Please see line 296-299.

***Modified text:***
line 280: "The concentrations of Ni, Cu, Zn, and Pb in were more variable than the crustal elements."

lines 287-291: "From the air mass back trajectories, the samples with high Ni (M2, M10), Cu (M1, M2, M4, M10) and Zn (M5) all were impacted by significant amount of air masses from the South Pacific Ocean and South America (Figure 5). The back trajectories of air masses of M7 didn't touch South America but the concentration of Zn in this sample was high. With the fact that aerosol Zn was found in both fine- and coarse-mode fractions (Table 2), both local sources and long-range transport may contribute to this element in the air."

lines 296-299: "In South America, high enrichment of Ni, Cu, Zn, and Pb in fine mode particles was reported to be primarily associated with vehicle emission, soil dust, and oil combustion (Artaxo et al., 1999; Jasan et al., 2009). Moreover, miming activities were suggested as an important source, especially in remote sites in South America (Carrasco and Préndez, 1991; Klumpp et al., 2000)."

Line 258 Please compare to other studies of aerosol Pb in the Southern Ocean.

***Response:*** We have added in the comparison with Bollhöfer et al. (2005) in Tasmania. Please see line 305.

***Modified text:*** line 305: "…and the annual average concentration of 11 pg m$^{-3}$ in Tasmania (Bollhöfer et al., 2005)."

Line263-264 "The low concentration of Pb observed in samples associated with trajectories that did not pass over Southern South America suggest that: : :"

*Response:* We have revised this sentence. Please see line 311-312.

*Modified text:* lines 311-312: "The low concentrations of Pb observed in samples associated with air masses that did not pass over Southern South America suggest that local anthropogenic emissions were negligible."

Line 265 ": : :long-range transported aerosol from Southern Hemispheric continental regions containing a mixture of anthropogenic and crustal emissions such a South America."

*Response:* We have revised this sentence. Please see lines 312-314.

*Modified text:* lines 312-314: "Thus the major source of non-crustal elements in aerosols over the study region may be long-range transport of aerosols from Southern Hemispheric continental regions containing a mixture of anthropogenic and crustal emissions."

Line 272 Do you mean K wasn't influenced by biomass burning. This should be mentioned before you use nss-K as a biomass burning tracer.

*Response:* At Palmer Station, the nss-$K^+$ was close to zero. It means the aerosol samples we collected were barely affected by biomass burning or any other sources, apart from sea spray, contribute $K^+$ to the atmosphere. We have added this interpretation in the "Mineral dust" part when we discuss the source of P (please see line 273-274).

*Modified text:* lines 273-274: "The calculated nss-K+ was indistinguishable from zero, suggesting that K+ in aerosol at Palmer Station was primarily derived from sea water, not from biomass burning through long-range transport."

Line 266-273 See modern day, coastal snow chemistry studies of Ca, Na and K derived from sea salt and sea ice to further back up the high abundance of marine derived elements at your site. Also, open the paragraph with a sentence to let the reader know you are discussing Ca, Na and K. End with a clear sentence stating the source(s).

*Response:* We have added a sentence in the beginning of this paragraph to show this part is about sea salt species. Please see line 316. We have also added a comparison of the Na/K and Na/Ca ratios with the snow samples collected at James Ross Island (Aristarain et al., 1982). We rephrased the last sentence to clearly suggest Ca and $K^+$ were primarily derived from sea salt aerosols. Please see lines 323-325.

*Modified text:*
line 316: "The concentrations of sea-salt species were much higher than the other trace elements (Table 1)."

lines 323-325: "Such results agree well with the Na/K (29.7) and Na/Ca (39.9) ratios

measured in snow samples collected at James Ross Island (Aristarain et al., 1982). Therefore, the Ca and K+ in aerosols were derived primarily from sea salt at Palmer Station."

Line 277 Consider renaming "elements from the continents" to "mineral dust"

**_Response:_** We have revised this section. We used k-means clustering to group all the elements based on their normalized particle size distributions. Please see lines 327-332.

**_Modified text:_** lines 327-332: "We applied k-means clustering to the full data set, and the results indicate that aerosol trace elements in the austral summer at Palmer Station can be classified into five groups based on their normalized particle size distributions (Figure 6), with each group showing a unique size distribution pattern: (1) crustal elements from crustal weathering and wind-induced resuspension of soil particles, (2) Al dominated by local minerals, (3) Pb from anthropogenic sources, as a result of long-range transport, (4) sea salt elements from the ocean, through bursting bubbles of seawater, and (5) P from local biogenic and soil resuspension."

L278 First mention of combustion. You previously ruled out biomass burning. Please state which elements are combustion products in the introduction or discussion section on sources.

**_Response:_** We have removed "combustion" and revised this section. Please see the first paragraph in section 3.3.

**_Modified text:_** Please see the revised section 3.3, lines 327-334.

Line 281 This group of elements all seem to have a coarse mode around the same size. Please include.

**_Response:_** We have revised this section. The elements showing single coarse mode distributions are discussed together in section 3.3.1, line 335-344. "This group of elements all have a coarse mode around the same size" is mentions in section 3.3.3, line 375-377.

**_Modified text:_**
lines 335-344: Please see section 3.3.1.

lines 375-377: "A single-mode Al particle size distribution peaked at around 4.4 µm which shared the same size with the primary mode of Ti, V, Mn, and Ce in sample M8 and M9, indicating the regional crustal emissions."

Line 281-282 Is this reference for long-range transported dust?

**_Response:_** This reference didn't specify long-range transported dust. We have removed

this citation.

Line 280-289 This section could be considerably improved. Some elements have a secondary mode in some samples but a single mode in other samples e.g. Al, V, Mn, Pb. Please discuss the episodic deposition from multiple sources and include in your discussion that these elements can have more than one source in a particular airmass. Please plot size distributions for all elements in Figure 6. I can't see Na and K. Include in your discussion a comparison with the literature to help attribute individual modes to sources. Some, but not all, references for Antarctic aerosol size distributions are below. I think the elements should be grouped differently according to the particle size distribution rather than the crustal vs non-crustal EF analysis. For example, crustal sources: P, Ti, Ca, and Ce have a single coarse mode distribution around XYZ um indicating. V and Mn have a bimodal distribution with a primary mode around XYZ um and secondary mode around ZXY um. Pb has a bimodal and sometimes trimodal distribution with a fine primary mode that is much finer than and mode in the other elements. Al, again, has a different distribution with a bimodal and sometimes trimodal distribution with mode similar to P, Ti, Ca, and Ce and an additional coarse model not seen in other elements. Also, Al exhibits a fine mode, similar to Mn and V, but only in one sample. Can you rule out local dust contamination for coarse Al concentrations? Could you compare the Al particle size distribution to mineral dust size distributions in Antarctic snow? A very interesting section could be developed about this comparison. Lachlan-Cope, T., Beddows, D.C., Brough, N., Jones, A.E., Harrison, R.M., Lupi, A., Jun Yoon, Y., Virkkula, A. and Dall'Osto, M., 2020. On the annual variability of Antarctic aerosol size distributions at Halley Research Station. Delmonte, B., Winton, H., Baroni, M., Baccolo, G., Hansson, M., Andersson, P., Baroni, C., Salvatore, M.C., Lanci, L. and Maggi, V., 2020. Holocene dust in East Antarctica: Provenance and variability in time and space. The Holocene, 30(4), pp.546-558. Fattori, I., Becagli, S., Bellandi, S., Castellano, E., Innocenti, M., Mannini, A., Severi, M., Vitale, V. and Udisti, R., 2005. Chemical composition and physical features of summer aerosol at Terra Nova Bay and Dome C, Antarctica. Journal of Environmental Monitoring, 7(12), pp.1265-1274. Virkkula, A., Teinilä, K., Hillamo, R., Kerminen, V.M., Saarikoski, S., Aurela, M., Koponen, I.K. and Kulmala, M., 2006. Chemical size distributions of boundary layer aerosol over the Atlantic Ocean and at an Antarctic site. Journal of Geophysical Research: Atmospheres, 111(D5). Fossum, K.N., Ovadnevaite, J., Ceburnis, D., Dall'Osto, M., Marullo, S., Bellacicco, M., Simó, R., Liu, D., Flynn, M., Zuend, A. and O'dowd, C., 2018. Summertime primary and secondary contributions to Southern Ocean cloud condensation nuclei. Scientific reports, 8(1), pp.1-14.

> ***Response:*** We would like to thank Dr. Winton for sharing these references with us. We have added a discussion about the potential contribution from long-range transport (anthropogenic and crustal) for Al, V, Mn, and Pb in line 351-359. We have added the plots for particle size distribution of Na and K in Figure 7.
>
> We have rewritten this section and grouped the elements by their particle size distributions. At the start of the revised section, we added a result of k-means clustering which classifies the particle size distributions of elements/ionic species into 5 clusters. The particle size distributions of all the elements and ionic species are discussed in 3 sections: single mode distribution, bimodal distribution, and trimodal distribution.

Special samples with unique size distribution are discussed with the air mass back trajectories in this section. At the end of the revised section, we also added a short discussion to compare Al size distribution in this study with the dust particle size distribution measured in snow, ice core, and aerosols in Antarctica. However, we can not rule out local dust contributions to the observed Al concentrations, and we have briefly mentioned this in the revised manuscript. Please see section 3.3.

*Modified text:* Please see all the changes in section 3.3 in lines 326-396.

Line 318-319 Note that estimates of wet deposition fluxes are even scarcer than dry deposition fluxes! Please state the limitations with this assumption but note it is likely the best estimate we have.

*Response:* We have added an explanation to show this estimate may have large uncertainty. Please see lines 412-413.

*Modified text:* lines 412-413: "Although precipitation varies in different regions, this is the best estimate we have for Antarctic regions."

Line 321 Dome C is a high elevation site and receives fine dust with a mode around 2-3 um.

*Response:* Thanks for sharing that information. We have added the comparison of dust particle size distribution in Dome C ice core samples (Delmonte et al., 2002) in discussion. Please see lines 392-394.

*Modified text:* lines 392-394: "Likewise, the dust particle size distribution in Dome C during the Last Glacial Maximum and the Holocene showed single-mode and peaked at approximately 2 - 3 μm in ice core samples (~3233 m above sea level) (Delmonte et al., 2002; Potenza et al., 2016)."

Line 305-340 Please add errors on these fluxes.

*Response:* Done. We have added errors in both text (line 401, 408, and 414) and revised Table 3.

Line 323-327 Suggest removing the comparison to values outside of the Southern Ocean and replacing with a discussion on why the flux is at the lower end of the JRI estimate.

*Response:* We didn't remove the comparison since it can give the readers a sense that the dust deposition flux at Antarctic Peninsula is extremely low.

There are several active dust sources reported in James Ross Island (Kavan et al., 2018), and we expect higher deposition fluxes close to the source region. In addition, our wet deposition was estimated based on an assumed percentage (40-60%), which could be

underestimated. This could also contribute to a low dust flux. We have briefly discussed this in the revised manuscript. Please see lines 418-420.

*Modified text:* lines 418-420: "Considering active dust sources were reported at James Ross Island (Kavan et al., 2018), sites close to James Ross Island are likely to receive higher dust deposition fluxes. In addition, without measurements of wet deposition, the total dust deposition flux in our study could be underestimated."

Line 332 Add reference for the 40 % assumption.

*Response:* Done. Please see line 430.

*Modified text:* line 430: "…assumed that dry deposition accounted for 40% of total deposition (Gao et al., 2003)…"

Line 331-333 What geographic area does the estimate represent?

*Response:* This estimate represents the West Antarctic Peninsula and its marginal seas. We have mentioned this in line 428.

*Modified text:* line 428: "To examine the potential importance of atmospheric dust input to the particulate elemental concentrations in surface waters of the West Antarctic Peninsula shelf region…"

Line 337-341 Could you estimate the lower bound as well and report both upper and lower bounds for summer? This information would be useful to include in the abstract too.

*Response:* Our current estimation is based on several assumptions that maximize the impact of atmospheric deposition. It gives us a sense that the atmospheric deposition only plays a minor role for suppling those trace elements to the west Antarctic Peninsula costal seawater. However, we don't have enough information to provide the upper and lower bounds.

Line 343 This conclusion omits previous discussion of anthropogenic emissions. Please keep a consistent message throughout the manuscript. What about biogenic emissions of P here and in the abstract?

*Response:* We have rewritten the conclusions and implications section (section 4). We include anthropogenic emission in the conclusion. Please see lines 444-445. A speculation of the P emission is given in lines 449-450.

*Modified text:*
lines 444-445: "…derived from (1) the regional crustal sources, which includes P-enriched soil resuspension, (2) remote anthropogenic emissions in South America , and

(3) seawater."

       lines 449-450: "We speculate that the modest enrichment of aerosol P over its crustal ratio to Al was caused by the resuspension of regional soils that are P-enriched as a result of the impact of nearby bird colonies."

Line 348-351 I don't see how you can make this claim without data prior to the commencement of shipping. There are regional differences in aerosol chemistry and dust fluxes around Antarctica. Perhaps rephrase indicating that future impacts of shipping and changes in the ice-free area have an unknown impact on aerosol chemistry in the region.

       ***Response:*** We have revised this sentence to suggest future study for the unknown impact of shipboard tourism and warming conditions on the aerosol chemistry in Antarctic Peninsula. Please see line 460-463.

       ***Modified text:*** lines 460-463: "The Antarctic Peninsula is experiencing rapid climate change, with the expansion of ice-free areas and more frequent shipboard tourism with unknown impacts on aerosol chemistry in this region. To quantify future changes in in the atmospheric position and the impact in this region, long-term atmospheric observations of aerosol chemical and physical properties along with coupled studies of the ocean-atmosphere interactions would be needed."

Fig 1 and Table 1: These look like copies of Fig. 1 and Table 1 in Gao et al. (2020). Please state if the figure is reproduced from this publication.

       ***Response:*** Yes. The Figure 1 and Table 1 are from Gao et al. (2020). We have referenced this paper.

Fig. 2: State what the label stands for. Please provide more information in the caption.

       ***Response:*** Done. We have added a description of the label in the figure caption.

Fig. 3: Please state the time period the samples represent in the caption.

       ***Response:*** Done. We have added the time periods in the figure caption.

Fig. 4: State that these are all land-based sampling sites except one cruise. Are these all total digestible concentrations? Why don't you add other data from cruises in the Southern Ocean?

       ***Response:*** We have added the information on either land-based or cruise in the figure caption. In this figure, we compared the previous observations of trace element concentrations in the coastal Antarctic region. We included the cruise from Zhongshan Station to Casey Station because it mainly focused on measuring the trace element concentrations in coastal East Antarctica. In this figure, not all the studies used the total

digestion method. Artaxo et al. (1990); Artaxo et al. (1992); Mazzera et al. (2001) and Préndez et al. (2009) used X-ray fluorescence to measure the total concentrations of trace elements. All the other studies applied acid digestion methods. We have added this information in the revised figure caption. Please see the revised Figure 4.

Fig 5: Why are you only showing air mass back trajectories of 4 samples? Please make the continent outline clearer as it is difficult to see under the air mass fetch regions.

**_Response:_** We have included the back trajectories for all 8 samples. Unfortunately, these figures were generated on NOAA's website and we cannot change the background. Please see the revised Figure 5.

Fig. 6: Please provide more information in the caption. Group the elements into the size distribution patterns discussed in the text. State what the axis labels stand for and include a reference for the notation in the methods. Are you plotting the number or volume particle size distribution? Please plot size distribution for ALL elements.

**_Response:_** In our study, we use the mass size distributions rather than number or volume particle size distributions. We have added a description about how the mass distribution was obtained in method in section 2.3.2, line 155-163. We have revised the figures. Now, the single mode particle size distributions are shown in Figure 7, and the bimodal/trimodal distributions are shown in Figure 8. We have added the figures for size distribution of $K^+$ and $Na^+$. The notation of normalized particles size distribution and reference have been added to Method. Please see section 2.3.2, line 156-161.

**_Modified text:_** We have added a new section in method. Please see section 2.3.2.

Tables 2-3: Could be moved to the supplement.

**_Response:_** We have moved Table 2 (LOD, %blank …) to the supplement. However, we believe that keeping Table 1 (sampling information) and Table 3 in the main body of the manuscript (concentrations of trace elements) is convenient to the readers.

**References**

[revised manuscript text omitted]